# *SpecPrune-VLA*: Accelerating Vision-Language-Action Models via Action-Aware Self-Speculative Pruning

**Hanzhen Wang** [* 1]  **Jiaming Xu** [* 1 2]  **Yushun Xiang** [1]  **Jiayi Pan** [1]  **Yongkang Zhou** [1 2]  **Yong-Lu Li** [1 2]  **Guohao Dai** [1 2 3]

## Abstract

Pruning is a typical acceleration technique for compute-bound models by removing computation on unimportant values. Recently, it has been applied to accelerate Vision-Language-Action (VLA) model inference. However, existing methods focus on local information from the current action step and ignore the global context of the model, leading to >20% success rate drop and limited speedup in some scenarios. In this paper, we point out **spatial-temporal consistency** in VLA tasks: input images in consecutive steps exhibit high similarity, and propose the key insight that token selection should combine local information with global context of the model. Based on this, we propose *SpecPrune-VLA*, a training-free, two-level pruning method with heuristic control. **(1) Action-level static pruning.** We leverage global history and local attention to statically reduce visual tokens per action. **(2) Layer-level dynamic pruning.** We prune tokens adaptively per layer based on layer-wise importance. **(3) Lightweight action-aware controller:** We classify actions as coarse- or fine-grained by the speed of the end effector and adjust pruning aggressiveness accordingly. Extensive experiments show that SpecPrune-VLA achieves up to 1.57× speedup in LIBERO simulation and 1.70× on real-world tasks, with negligible success rate degradation.

## 1. Introduction

Vision-Language-Action (VLA) models, built upon large language models (LLMs), have gained attention for their

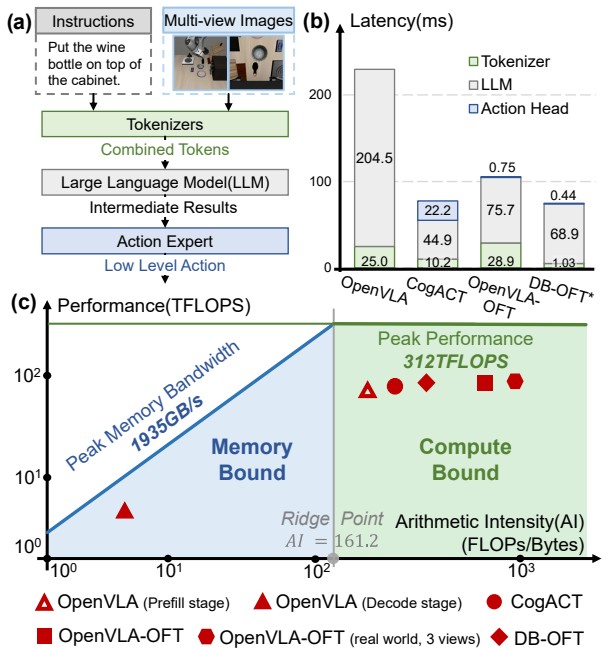

*Figure 1.* (a) The mainstream inference dataflow of VLA models. (b) Latency breakdown in four typical VLA models during each action generation. * In DB-OFT, we report the time of LLM backbone and Action Expert per denoising step, with the tokenizer's single forward pass time averaged evenly across the 10 steps. (c) The practical arithmetic intensity of four models in different cases in the roofline model of NVIDIA A800 GPU.

ability to interpret multimodal inputs and generate robotic actions. Models like RT-1 (Brohan et al., 2022) and Open-VLA (Kim et al., 2024) demonstrate strong cross-task generalization and instruction-following capabilities from real-world robot data. Follow-up works (Team et al., 2024; Li et al., 2024b; Black et al., 2024; Kim et al., 2025) are further proposed for real-time performance improvement. As shown in Figure 1(a), VLA inference typically involves: (1) tokenizers for encoding multimodal inputs, (2) an LLM backbone for processing multimodal tokens and generating intermediate outputs, and (3) an action head for producing low-level actions. We profile four representative models, auto-regressive model OpenVLA (Kim et al., 2024), model with a diffusion action expert CogACT (Li et al., 2024b), model with a linear action expert OpenVLA-OFT (Kim

---

[*]Equal contribution  [1]Shanghai Jiao Tong University  [2]SII  [3]Infinigence-AI. Correspondence to: Guohao Dai <daiguohao@sjtu.edu.cn>.

*Proceedings of the 43^rd International Conference on Machine Learning*, Seoul, South Korea. PMLR 306, 2026. Copyright 2026 by the author(s).

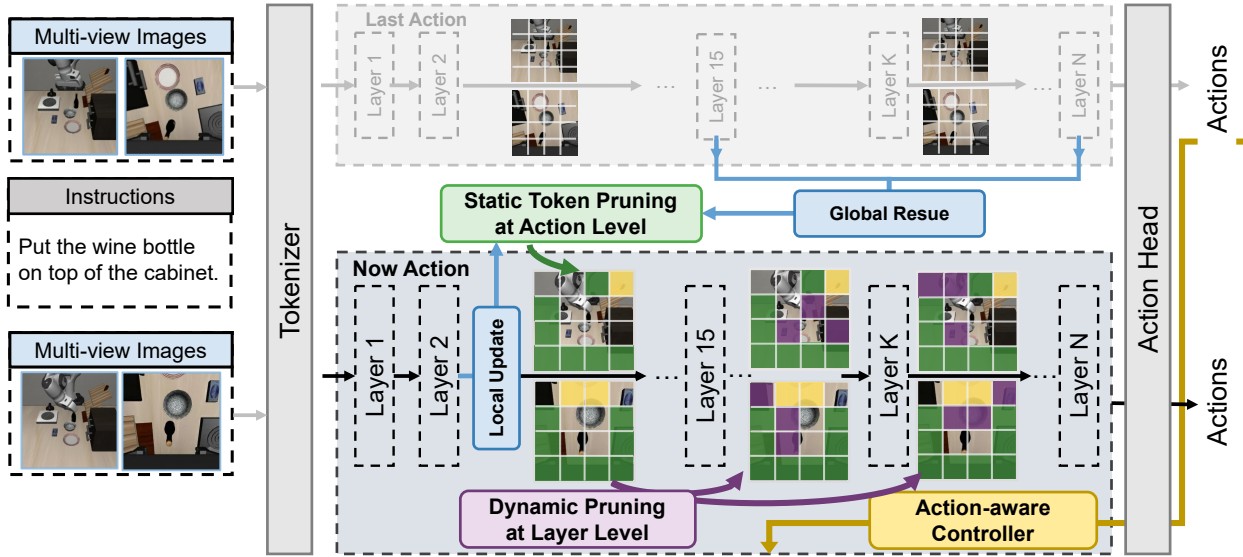

*Figure 2.* Overview of SpecPrune-VLA. We prune the visual tokens with global and local information with a lightweight action-aware controller.

et al., 2025) and diffusion-based model Dexbotic-OFT (DB-OFT) (Xie et al., 2025) in Figure 1(b) and reveal that the LLM is the critical inference bottleneck ($> 70\%$ of the end-to-end latency in most models). Therefore, most works target LLM acceleration via various acceleration methods (Xu et al., 2025c; Hong et al., 2024; Park et al., 2024; Yue et al., 2024; Xu et al., 2025b). However, they largely overlook VLA-specific computation patterns, limiting their effectiveness. Nowadays, the latest VLA models (*e.g.*, OpenVLA-OFT and DB-OFT) adopt the single-step paradigm that the model directly predicts a sequence of low-level actions through a single LLM forward (*i.e.*, only prefill phase) with hundreds of multimodal tokens. As a result, from the perspective of arithmetic intensity (*i.e.*, the amount of computation per byte) in the hardware roofline model, the VLA inference is primarily compute-bound, as shown in Figure 1(c), where latency mainly arises from the amount of computation rather than memory access.

Pruning is a typical acceleration method for compute-bound problems by effectively reducing the computation (Zhou et al., 2024). However, existing token-pruning methods in VLA models (Yang et al., 2025; Li et al., 2025) only consider local information (*e.g.*, early layer results in current action generation) and ignore global information (attention of deeper layers) across the whole model, leading to either $>$ 20% success rate loss or limited speedup in some scenarios.

In this paper, we point out that input images in consecutive action generations exhibit high similarity due to the short temporal intervals between them. Therefore, we consider that the global information from previous inference steps can be leveraged for more reliable and efficient token pruning. Based on the above insight, we propose SpecPrune-VLA, an acceleration method for Vision-Language-Action

models through action-aware self-speculative pruning. The techniques of SpecPrune-VLA can be summarized in three points as follows.

**(1) Action-level static pruning.** Based on the insight, we point out that tokens between consecutive action generation are largely overlapped (*e.g.*, the background in the environment), leading to significant information redundancy. Therefore, we reuse the attention information of the global model(the middle layer and deep layer) from the last generation to prune globally unimportant tokens. Then we enhance it with dynamic elements and current task-relevant tokens by speed-based frame comparison and self-speculative token selection. By fusing tokens selected from local and global levels, we can prune 60% to 70% visual tokens at the beginning of LLM forward.

**(2) Layer-level dynamic pruning.** As the input features propagate through the LLM backbone, the local context of each token is progressively enriched by deeper layers. Therefore, we introduce layer-wise pruning by dynamically updating tokens' importance scores and re-evaluating token importance at different depths. This allows the model to adaptively refine computation focus and remove redundant tokens as contextual understanding matures, reducing extra 20% computation.

**(3) Lightweight action-aware Controller.** We propose that not all actions are equally sensitive to token pruning. Therefore, we categorize actions into coarse-grained (*e.g.*, large translations) and fine-grained (*e.g.*, grasping) types and design a controller. It determines action granularity based on the speed of the end-effector and adaptively adjusts the pruning aggressiveness with negligible overhead, enabling a robust trade-off between speed and accuracy across diverse

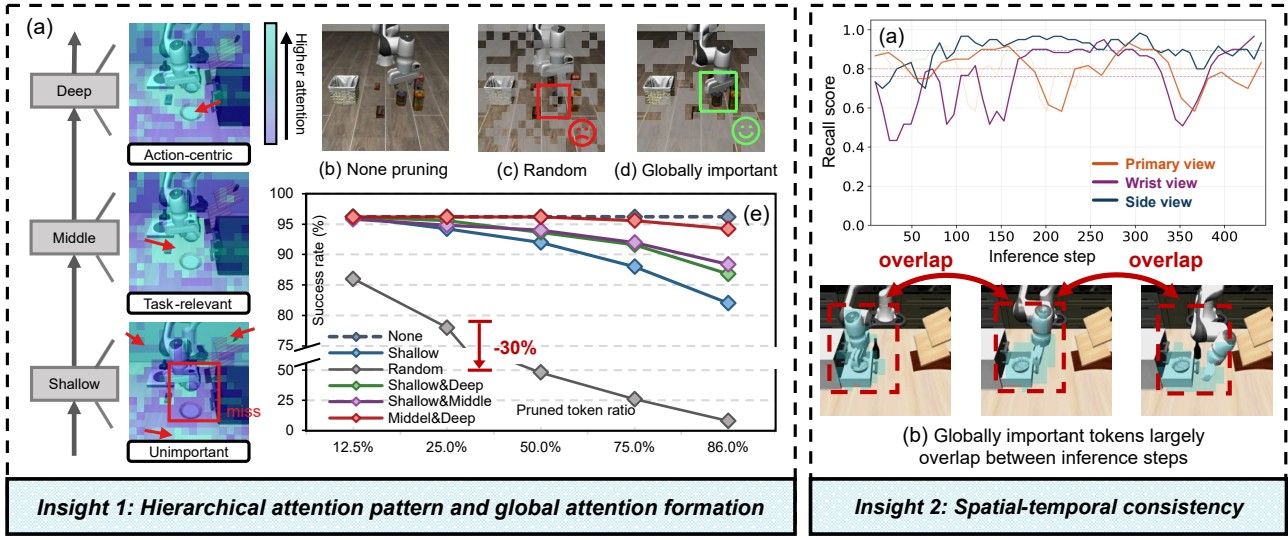

*Figure 3.* Insight 1: (a) Layers of different depth focus on different information. (b)(c)(d) In pick and place task, random pruning causes important tokens missed out and pruning based on global attention retain important tokens and ensure success rate. (e) The performance of different guiding layer selection. Insight 2: (a) The recall score of Top-30 globally important token sets of consecutive inference remains high in different camera view. (b) The visualization of globally important tokens between consecutive steps.

robotic tasks.

We implement our method on models with three different popular architectures: OpenVLA-OFT, DB-OFT and CogACT. Our method lead to consistently high speedup across all models and achieves up to $1.57\times$ speedup compared with OpenVLA-OFT on the LIBERO simulation benchmark, and $1.70\times$ speedup in real-world tasks, with negligible success rate loss.

## 2. Related Works

### 2.1. Vision-Language-Action (VLA) Models

VLA models are typically LLM-based (Zitkovich et al., 2023; Liu et al., 2023b), fine-tuned on large-scale simulated (Liu et al., 2023a) and real-world (O'Neill et al., 2024) robotic datasets. They process multimodal inputs (e.g., images + text) to generate low-level robotic actions. Continuous action spaces are preferred for higher manipulation accuracy (Liu et al., 2025), often decoded via lightweight MLPs or diffusion heads (Liu et al., 2024; Wen et al., 2025). To ensure high control frequency and temporal coherence, modern VLAs adopt ACT (Zhao et al., 2023), diffusion models (Peebles & Xie, 2023), or parallel decoding for chunked action generation (Li et al., 2024a).

### 2.2. Token-level Acceleration for VLA model

Recent works explore token caching or pruning. VLA-Cache (Xu et al., 2025c) reuses cached key-value pairs from unimportant tokens, but only reduces 17–25% of total FLOPs and introduces additional GPU memory access overhead. EfficientVLA (Yang et al., 2025) prunes visual

tokens using single-layer attention heuristics and supplements with diverse patches — yet this risks introducing task-irrelevant content and lacks global context awareness. SP-VLA (Li et al., 2025) retains tokens with high vision encoder saliency to preserve spatial-semantic structure, but still fails to filter semantically redundant tokens, leaving unnecessary computation.

### 2.3. Self-Speculative Decoding and Lightweight Predictors

Unlike standard speculative decoding (Leviathan et al., 2023) requiring a separate draft model, LayerSkip (Elhoushi et al., 2024) uses early layers of the same model for drafting and deeper layers for verification, reducing memory and latency. Separately, SpecEE (Xu et al., 2025b) and SpeContext (Xu et al., 2025a) employ a lightweight predictor to filter low-probability tokens and dynamically load KV cache based on attention score, significantly lowering decoding cost.

## 3. Key Insights

### 3.1. What Really Matters in the Image

We systematically study which image components are critical for the model. As shown in Figure 3, Insight 1(a), image-to-text attention patterns evolve across layers. In the task "put the bowl on the plate", shallow layers attend broadly, including background and irrelevant regions (e.g., table) but miss important object (e.g. bowl and plate); middle layers focus on semantically relevant objects that inform task understanding even though they may not involve with

the action (e.g., cabinet); deep layers focus on action-centric tokens directly involved in execution (e.g. plate).

To assess the value of this hierarchical attention, we conduct a post-hoc token pruning experiment. Using attention scores between image to text (Eq. 2) — a commonly used importance proxy (Zhang et al., 2024b;a; Ye et al., 2025)—we identify layer-wise important tokens. Actions are first generated without execution, then tokens are pruned based on attention scores, and actions are regenerated from the compressed input and executed.

Results (Figure 3, Insight 1(e)) show that random pruning maintains performance only up to 12.5% sparsity, beyond which accuracy drops sharply, which indicates redundancy exists but also informed pruning guidance is needed. Pruning guided by shallow layers performs poorly under high sparsity (>10% drop), as they capture irrelevant, redundant information (e.g. table texture and background). In contrast, strategies combining mid and deep layers achieve superior robustness, with minimal degradation even at 86% pruning. This demonstrates that fusing task-relevant and action-centric representations provides a reliable signal for efficient model compression.

## 3.2. Information largely overlaps in images of consecutive inference

***Globally important tokens*** needs to be retained to ensure accuracy. It is challenging to identify these tokens before the whole model completes current inference. Current methods, such as (Li et al., 2025; Yang et al., 2025), utilize local information such as the attention score of one LLM layer or the vision encoder. However, they didn't consider global information from the whole model and are thus not reliable. In this paper, we emphasize that in VLA models, the overall task goal remains constant and a large proportion of the visual scene remains unchanged across consecutive inferences due to the minimal time change. Therefore, tokens identified as globally important in the previous generation are likely to remain important in the current step as shown in Figure 3, Insight 2(b). We call this spatial-temporal consistency.

To quantify this consistency in token importance, we define the ***Recall of Important Tokens***, which measures the overlap between the globally important token set at the previous step $V_{t-1}$ and the current set $V_t$, normalized by the size of the current set. Formally, it is expressed as:

$$\text{Recall}(V_{t-1}, V_t) = \frac{|V_{t-1} \cap V_t|}{|V_t|}. \qquad (1)$$

As shown in Figure 3, Insight 2(a), we observe that this recall reaches an average of 75% -88% from different viewpoints throughout the task execution, indicating strong temporal persistence in token relevance. This temporal consistency inspires us to reuse the global attention scores across

time.

# 4. Action-level Static Token Pruning

## 4.1. Method

### 4.1.1. PRUNING BASED ON GLOBAL INFORMATION

As illustrated in Section 3.1, in cross-attention layers where visual tokens serve as queries and text tokens as keys, a high attention weight from a visual token $V_i$ to task-instruction (text) tokens indicates that the visual token is important.

Given a unified input sequence containing $n$ visual tokens $\mathcal{V} = \{V_1, \ldots, V_n\}$ and $m$ textual tokens $\mathcal{T} = \{t_1, \ldots, t_m\}$ (e.g., task instructions) processed by a single Transformer layer, we identify task-relevant visual tokens by measuring how actively each visual token *attends to* the textual tokens.

Formally, let $A_l^h \in \mathbb{R}^{(n+m) \times (n+m)}$ denote the attention matrix of head $h$ at layer $l$, where the entry $A_l^h(i, j)$ represents the attention weight when the $i$-th token serves as query and the $j$-th token serves as key. For visual token $V_i$ and textual token $t_j$, we extract the image-to-text attention weight as: $A_l^h(p_i, q_j)$. The task-relevance score of $V_i$ in layer $l$ is defined as the average attention it allocates to *all* instruction tokens across all heads:

$$\text{Score}_l(V_i) = \frac{1}{H \cdot m} \sum_{h=1}^{H} \sum_{j=1}^{m} A_l^h(V_i, t_j), \qquad (2)$$

Then we define $V_{global}$ as the set of the top-$K_{global}$ visual tokens with the highest such attention scores from the middle and deep layers (we choose the 15th and 32nd layers) in the prior inference step. Based on our key insight that global information exhibits temporal consistency across consecutive actions, we retain $V_{global}$ in the current step.

### 4.1.2. SUPPLEMENTATION OF DYNAMIC TOKENS

Visual tokens undergoing significant changes between inference steps cannot be reliably pruned using global information from the prior step. To preserve up-to-date content, we explicitly retain these *dynamic* tokens during static pruning. Given frames $I_m$ and $I_n$, we partition each into $N \times N$ patches according to the token size. Let $\mathbf{P}_t^{i,j}$ denote the feature vector of patch $(i, j)$ in frame $I_t$. The cosine similarity between corresponding patches is:

$$\text{Sim}(\mathbf{P}_m^{i,j}, \mathbf{P}_n^{i,j}) = \frac{\mathbf{P}_m^{i,j} \cdot \mathbf{P}_n^{i,j}}{\|\mathbf{P}_m^{i,j}\|_2 \|\mathbf{P}_n^{i,j}\|_2}.$$

To identify dynamic tokens, we first filter patches with similarity scores below a threshold $\tau$, then select the top-$k$ patches with the lowest similarity scores from the remaining candidates. Formally, let $\mathcal{P}_n = \{\mathbf{P}_n^{i,j} \mid 1 \leq i, j \leq N\}$ be the set of all patches in frame $I_n$. We define the candidate

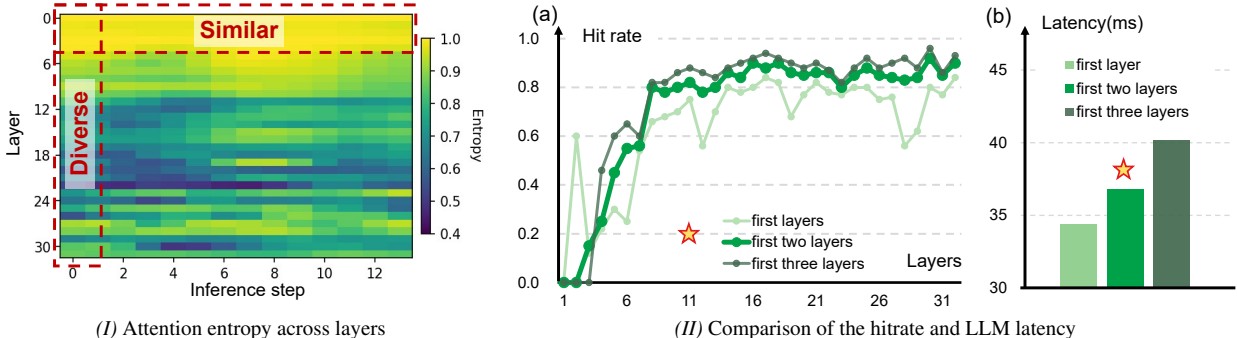

*(I) Attention entropy across layers*  *(II) Comparison of the hitrate and LLM latency*

*Figure 4.* (I) Attention entropy differs across layers, but are similar throughout the task. (II) (a) Comparison of the hitrate between leveraging the first one, two, and three layers. (b) LLM latency comparison between leveraging the first one, two, and three layers.

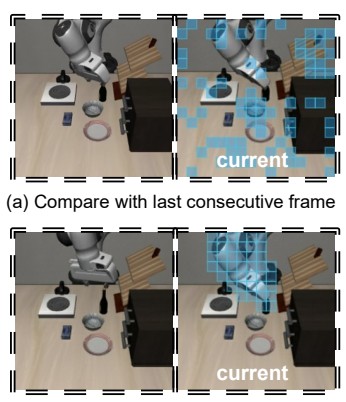

(a) Compare with last consecutive frame

(b) Compare with frame chosen by velocity

*Figure 5.* Adaptive selection strategy for frame comparison.

dynamic patches as those with significant changes:

$$\mathcal{C}_n = \left\{ \mathbf{P}_n^{i,j} \in \mathcal{P}_n \mid \mathrm{Sim}(\mathbf{P}_m^{i,j}, \mathbf{P}_n^{i,j}) < \tau \right\}. \tag{3}$$

The most dynamic $K_{dynamic}$ tokens are then given by:

$$V_{dynamic} = \text{Low-}K_{dynamic}\left( \left\{ \mathrm{Sim}_{i,j} \mid \mathbf{P}_t^{i,j} \in \mathcal{C}_t \right\} \right), \tag{4}$$

Additionally, as shown in Figure 5(a), directly comparing adjacent frames can yield inaccurate results due to camera noise and light changes, especially in real-world scenarios. Therefore, we propose a ***velocity-based frame sampling*** strategy. This method selects a historical reference frame that is $T$ frames before the current one, where $T$ is calculated as: $T = \lfloor b + k \cdot v \rfloor + 4$. Here, $k = -1$ and $b = 7$ are constants based on experimental results. $k$ inversely relates speed $v$ to $T$, while $b$ adjusts the baseline value of $T$. The translational speed $v$ is discussed in Section 6.

#### 4.1.3. PRUNING BASED ON LOCAL INFORMATION

Due to changing sub-goals and images, we need to incorporate information of current generation by analyzing attention-based importance using Eq. 2. We observed that 80%–90% of top-k important tokens from the first two layers reappear in the final layer's top-k (Figure 4II, k=30),

indicating early-layer attention provides reliable guidance for token selection. Besides, the first layer alone shows a low hit rate and adding the third layer provides marginal gains with extra latency. Considering precision and efficiency, we use the first two layers for speculation to filter current important tokens.

In each layer, we select the $K_{local}$ visual tokens with the highest attention scores to form a candidate set $V_{(1)}$ and $V_{(2)}$ respectively, and take the union of these two sets as the local information representation: $V_{local} = V_{(1)} \cup V_{(2)}$. Finally, all the retained token set is: $V_{retain} = V_{global} \cup V_{dynamic} \cup V_{local}$.

## 5. Layer-level Dynamic Token Pruning

To preserve the most important tokens in layers, we propose dynamic importance scoring mechanism that leverages attention scores and layer confidence across LLM layers to prune tokens within layers.

### 5.1. Importance Score Formulation

The token importance score is initialized for the remaining visual tokens after static token pruning and subsequently updated in the target transformer layers. The importance score $s_i^{(l)}$ takes into account both the relative importance weight of tokens and the layer contribution:

$$s_i^{(l)} = \omega_{\text{rank},i}^{(l)} \times \omega_{\text{conf}}^{(l)} \tag{5}$$

where $\omega_{\text{rank},i}^{(l)}$ denotes *rank-based weight* reflecting the token's relative importance in attention ranking, $\omega_{\text{conf}}^{(l)}$ denotes *layer confidence score* measuring the layer's reliability

**Rank-based Weight**  For each attention head, visual tokens are ranked based on their image-to-text attention scores in (2). To emphasize the contribution of the most important tokens while maintaining a smooth decay in influence, we introduce a rank-based weighting scheme. This weight is

defined as:

$$\omega_{\text{rank},i}^{(l)} = \frac{\sigma(-k \cdot \text{rank}_i^{(l)})}{\sum_j \sigma(-k \cdot \text{rank}_j^{(l)})} \quad (6)$$

where $\text{rank}_i^{(l)}$ is the attention ranking of token $t_i$ in layer $l$ and $\sigma(x)$ denotes the sigmoid function, which amplifies the differences between token rankings by mapping them to a smooth range, ensuring that higher-ranked tokens receive significantly more emphasis.

**Layer Confidence Score**  Inspired by (Zhang et al., 2025b), in Transformer layers, high attention entropy indicates a dispersed attention distribution, where the model fails to concentrate on salient tokens. As shown in Figure 4I, we observe that the attention entropy across layers in our VLA model varies significantly with depth, suggesting that different layers contribute unequally to identifying globally important information.

We posit that layers with low entropy, focused attention are more reliable for token importance estimation. Let $A_{ij}^{(l)}$ denote the attention weight from text query token $i$ to image key token $j$ in layer $l$ of the image-to-text attention. The average attention entropy is computed as:

$$\bar{H}^{(l)} = -\frac{1}{N} \sum_{i=1}^{N} \sum_{j=1}^{M} A_{ij}^{(l)} \log A_{ij}^{(l)}, \quad (7)$$

where $N$ and $M$ are the numbers of query and key tokens. We then compute the layer confidence score $\omega_{\text{conf}}^{(l)}$ as:

$$\omega_{\text{conf}}^{(l)} = \frac{1}{\bar{H}^{(l)} + \epsilon}, \quad (8)$$

with $\epsilon > 0$ for numerical stability. Lower entropy corresponds to higher confidence, reflecting more focused and semantically grounded attention. This value is computed in the first inference step and reused thereafter due to high inter-step similarity (see Appendix A.4).

## 5.2. Dynamic Updating Mechanism

The final importance score $S_i$ for each token $t_i$ is maintained through an exponential moving average across layers:

$$S_i^{(l)} = (1 - \beta) \cdot S_i^{(l-1)} + \beta \cdot s_i^{(l)} \quad (9)$$

where $\beta$ is the learning rate controlling the update speed, set to 0.2, and $S_i^{(0)} = 0$ for initialization. For layers in update layer set, we prune 10% tokens with the lowest score.

# 6. Lightweight Action-aware Controller

## 6.1. Observation and Insight

Empirically, aggressive token pruning leads to a drop in success rate. Frame-by-frame observation reveals that failures predominantly occurred during object-contact phases,

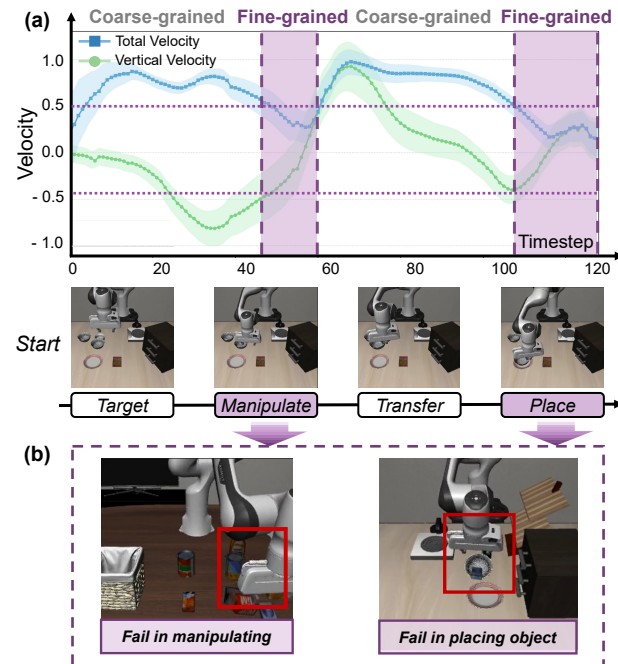

*Figure 6.* (a) The task process consists of four stages, categorized into coarse- and fine-grained actions based on velocity. (b) Typical failures in fine-grained stages.

such as manipulation or placement (Figure 6(b)) where even minor errors cause task failure. The task merely fails when those actions are successfully executed. This highlights that task success critically depends on *fine-grained* actions, which demand high precision and are sensitive to pruning. In contrast, *coarse-grained* actions (*e.g.* moving to a general location) tolerate more approximation. Specifically, when the robot approaches an object, fine-grained control is essential for stable contact and successful execution. Thus, action granularity dictates the required level of visual fidelity and inference precision.

Inspired by this, we propose an action-aware pruning strategy: by detecting whether a step requires fine or coarse control, our method preserves more tokens during fine-grained phases and prunes more aggressively during coarse-grained ones, improving both efficiency and success rate.

## 6.2. Method

Since actions span a fixed duration, end-effector velocity is measured as displacement per step. As all training data are normalized prior to model input, the output displacements $(\Delta x, \Delta y, \Delta z)$ and angular changes $(\Delta\alpha, \Delta\beta, \Delta\gamma)$ are inherently in normalized form. This normalization ensures that velocity magnitudes lie in a consistent range across tasks and platforms, making our method generalizable and independent of specific robot kinematics or environmental scaling (Other settings result in Appendix A.1.3). The

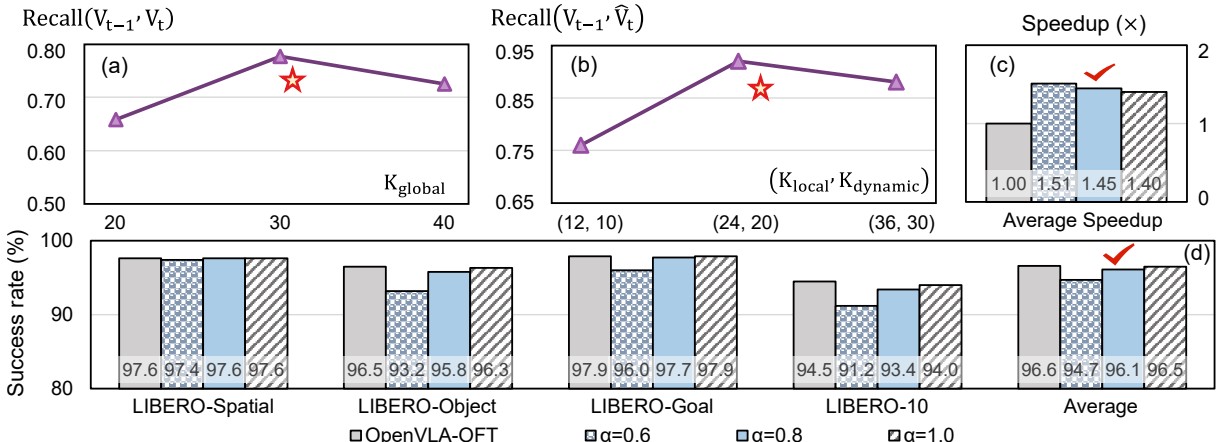

*Figure 7.* (a) and (b) Ablation study on base K value. K values are chosen to maximize the Recall rate. (c) and (d) Ablation study on prune rate.

translational and rotational velocities are computed as:

$$v_t = \sqrt{(\Delta x)^2 + (\Delta y)^2 + (\Delta z)^2}, \qquad (10)$$

$$v_r = \sqrt{(\Delta \alpha)^2 + (\Delta \beta)^2 + (\Delta \gamma)^2}, \qquad (11)$$

Analysis of trajectory data in Figure 6 reveals the distribution of bimodal velocity distributions between coarse and fine-grained phases. In coarse-grained phase, the overall velocity is high. During fine-grained phase, the translational and rotational velocity is typically slow, with non-positive $z$-axis displacement $\Delta z$. From this, we empirically identify thresholds: $v_t^{th}, v_r^{th}$. The system enters precise mode when $v_t < v_t^{th}$, $v_r < v_r^{th}$, and $\Delta z \leq 0$, and exits upon exceeding $v_t^{th}$ or $v_r^{th}$ (*e.g.*, during lifting). This adaptive control balances accuracy and efficiency (Alg. 1, Appendix).

## 7. Experiment

### 7.1. Experimental Settings

**Simulation Benchmarks and Platforms**   We conduct evaluations on both the LIBERO (Liu et al., 2023a) and SimplerEnv (Li et al., 2024c) simulation benchmark (Liu et al., 2023a), and in the real world. In LIBERO benchmark, we employ eight task suites, LIBERO-Spatial, LIBERO-Object, LIBERO-Goal, and LIBERO-Long, to evaluate the model's capabilities in spatial reasoning, object understanding, goal-directed planning and execution, and long-horizon task completion. In SimplerEnv, we include four visual matching tasks: Put Spoon on Towel, Put Carrot on Plate, Stack Cube and Put Eggplant in Yellow Basket. This setting minimizes domain shifts between simulation and reality, thereby replicating real-world tasks with high accuracy. All main experiments are conducted on a Linux workstation with an NVIDIA A800-80GB GPU.

**Baselines**   We select OpenVLA-OFT and DB-OFT as our target models . OpenVLA-OFT is VLM-based and DB-OFT

is Diffusion-based. OpenVLA-OFT utilizes a four-layer MLP as an action expert to generate continuous actions. DB-OFT runs the LLM backbone and action expert for 10 denoising steps, where the action expert processes the hidden feature of the LLM backbone and generates continuous actions. The visual and text tokens are fed into model at every denoising step. OpenVLA-OFT receives two-view images: the primary view and the wrist view while DB-OFT only receives primary view image. We also consider five optimization methods: SparseVLM (Zhang et al., 2024b), a framework that adaptively sparsifies less important visual tokens and recycles their information to minimize performance loss, DivPrune (Alvar et al., 2025) selects diverse visual tokens to preserve information and maintain accuracy. FastV (Chen et al., 2024) prunes redundant visual tokens early in the model based on observed sparse attention patterns. VLA-Cache (Xu et al., 2025c) leverages image similarity to cache features across time steps and EfficientVLA (Yang et al., 2025), a visual and structural pruning approach for VLA models. VLA-Cache and Efficient-VLA is originally designed for VLM-based VLA models, therefore, we only implement them on OpenVLA-OFT.

### 7.2. Parameter Setup

The base K values are chosen to maximize the temporal consistency of important information, measured by the ***Recall of Important Tokens***(Eq.(1)) as mentioned in our motivation. In Figure 7, for the $K_{global}$, we evaluated value of 20, 30, 40 and observed the average recall rate $\text{Recall}(V_{t-1}, V_t) = |V_{t-1} \cap V_t|/|V_t|$ is the highest when $K_{global}$ equals 30. For $K_{local}$ and $K_{dynamic}$, we tested several pairs: (12, 15), (24, 20), (36, 30). The token set $\hat{V}_t$ ($V_t$ supplemented by local and dynamic tokens), reaches the highest recall rate $\text{Recall}(V_{t-1}, \hat{V}_t) = |V_{t-1} \cap \hat{V}_t|/|\hat{V}_t|$ when the pair equals (24, 20). Therefore, we set the base value $K_{global}$=30, $K_{local}$=24 and $K_{dynamic}$=20.

*Table 1.* Performance Evaluation (Success Rate and Average Speedup)

| LIBERO benchmark | Success Rate (%) | | | | Avg. Speedup | Avg. SR(%) | FLOPs |
|---|---|---|---|---|---|---|---|
| Method | Spatial | Object | Goal | Long | | | |
| OpenVLA-OFT | 97.6% | 96.5% | 97.9% | 94.5% | 1.00× | 96.6% | 100% |
| + FastV [ECCV24] | 94.6% | 95.8% | 94.0% | 88.8% | 1.44× | 93.3% | 57% |
| + DivPrune [CVPR25] | 92.4% | 91.2% | 89.0% | 84.8% | 1.46× | 89.4% | 54% |
| + SparseVLM [ICML25] | 96.8% | 94.2% | 97.6% | 93.6% | 1.28× | 95.6% | 77% |
| + VLA-Cache [NIPS25] | 99.0% | 97.7% | 97.4% | 93.6% | 1.07× | 96.9% | 83% |
| + EfficientVLA [NIPS25] | 96.5% | 91.1% | 96.0% | 72.1% | 1.52× | 88.9% | 35% |
| **+ Ours** ($\alpha$=0.8) | 97.4% | 95.8% | 97.7% | 93.4% | 1.46× | 96.1% | 43% |
| SimplerEnv benchmark | Success Rate (%) | | | | Avg. Speedup | Avg. SR(%) | FLOPs |
| Method | PutCarrot | PutSpoon | StackCube | PutEggplant | | | |
| DB-OFT | 64.1% | 89.2% | 30.8% | 97.5% | 1.00× | 70.4% | 100% |
| + FastV [ECCV24] | 59.8% | 83.5% | 24.0% | 94.0% | 1.40× | 65.3% | 56% |
| + DivPrune [CVPR25] | 60.7% | 85.0% | 24.8% | 96.3% | 1.43× | 66.7% | 54% |
| + SparseVLM [ICML25] | 62.2% | 86.4% | 26.8% | 96.3% | 1.28× | 67.9% | 75% |
| **+ Ours** ($\alpha$=0.8) | 63.8% | 88.7% | 30.0% | 98.0% | 1.44× | 70.1% | 42% |

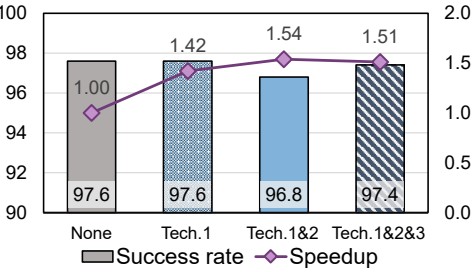

*Figure 8.* Ablation study of three techniques in LIBERO-spatial

## 7.3. Design Space Exploration

We use the prune ratio $\alpha$ to adjust the overall K values by scaling $K_i = \alpha \cdot K_i$ (The detail setup is in A.6.2) to control the aggressiveness of our pruning. Therefore we conduct a design space exploration to explore the impact of different prune ratios. As the Figure 7 shows, the smaller the prune ratio, the more tokens are pruned, leading to a drop in success rate and a rise in speedup. To balance accuracy and speed, we set the prune ratio $\alpha$ to 0.8 for a universal setup.

## 7.4. Evaluation on Speedup and Success Rate

Table 1 shows the end-to-end evaluation on success rate (SR), latency and speedup on four LIBERO and SimplerEnv task suits, respectively. SpecPrune-VLA reduces FLOPs by 57% and 58% and achieves an average speedup of 1.46× and 1.34× with negligible loss ($< 0.7\%$) in success rate separately compared to baseline model OpenVLA-OFT and DB-OFT. SparseVLM yields limited speedup and degraded SR, as its pruning, recycling, and merging strategy only minimally reduces computation and is not suited for precise action generation. Compared to FastV and DivPrune (both retaining similar token counts with our method for fairness), our method achieves comparable speedup but superior SR. FastV prunes tokens using early-layer attention without

training, ignoring global context; DivPrune maximizes feature diversity but neglects task-relevant token importance in VLA models. VLA-Cache maintains high success rate by mitigating noise and improving motion continuity through caching, yet achieves limited speedup due to limited computation reduction (17%) and GPU memory access overhead. EfficientVLA (L=28, T=112) attains a higher speedup by skipping layers and aggressively pruning tokens, but suffers notable SR drops in certain scenarios by compromising critical action-related information in hidden states.

## 7.5. Ablation Study

### 7.5.1. ABLATION ON THREE TECHNIQUES

To evaluate the effectiveness of our proposed method, we conducted an ablation study on the LIBERO-Spatial task suit. (Figure 8) Our full model achieves a success rate(SR) of 97.4%, comparable to the baseline (97.6%), and it significantly reduces latency from 109ms to 72.3ms, resulting in a speedup of 1.51× compared to OpenVLA-OFT. This demonstrates the efficiency gains of our approach while maintaining competitive accuracy. The ablation study further highlights the importance of each component: Static (Tech 1) and Dynamic (Tech 2) pruning slightly affect the SR (96.8%) but reduce latency to 70.8ms, indicating that pruning contributes to the overall latency reduction. The introduction of the action-aware controller increases the success rate and causes negligible latency (1.5ms). This suggests that the controller plays a crucial role in maintaining high accuracy.

### 7.5.2. ABLATION ON GLOBAL ATTENTION REUSE

We compare recall rate and success rate of pruning with and without global attention reuse in static pruning stage. For fair comparison, we set $K_{local}$= 54 when without global

*Table 2.* Performance comparison on real-world robot tasks.

| Real world task | Success Rate (%) | | | | Avg. | Avg. |
| Method | Pick&Place | TransferTube | PickUpCup | MultiCubeTask | Latency (ms) | Speedup |
|---|---|---|---|---|---|---|
| OpenVLA-OFT | 96.7% | 85.0% | 91.7% | 95.0% | 187.7 | 1.00× |
| Ours | 96.7% | 82.0% | 90.0% | 95.0% | 110.2 | 1.70× |

attention reuse, otherwise, we set $K_{local} = 24$, $K_{global} = 30$. As shown in Table 3, incorporating global context yields better token recall and higher success rates, indicating that prior-step attention provides valuable, task-relevant and action-centric guidance. This supports our key insight: spatio-temporal coherence in robotic tasks makes previous-step global attention a reliable prior for the current inference step.

### 7.5.3. ABLATION ON ENTROPY-BASED LAYER WEIGHT

We compare recall rate and success rate of pruning with and without entropy-based layer weighting in dynamic pruning stage in Table 4. From the perspective of recall rate, average weighting utilizes uncertain information from high-entropy layers and mistakenly prunes globally important tokens, thus achieving poorer success rate.

*Table 3.* Ablation on global attention

| Method | Recall (%) | LIBERO SR (%) |
|---|---|---|
| w/ global attn. | 92% | 96.1% |
| w/o global attn. | 84% | 93.4% |

*Table 4.* Ablation on entropy-based layer weighting

| Method | Recall (%) | LIBERO SR (%) |
|---|---|---|
| Entropy-based | 88% | 96.1% |
| Average | 66% | 92.0% |

### 7.6. Evaluation on Various Computing Platforms

To validate the applicability of our method on different devices, we conduct experiments on NVIDIA GeForce RTX 3090. As illustrated in Figure 9, our method achieves an average speedup of **2.09×** in LLM inference time and **1.57×** in end-to-end latency. The results consistently demonstrate improved inference efficiency, underscoring the scalability and effectiveness of our approach under diverse computational conditions.

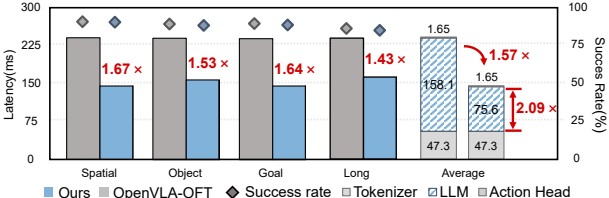

*Figure 9.* Extended evaluation on NVIDIA 3090 GPU

### 7.7. Evaluation on real robot

**Experimantal settings**   In this section, we evaluate the real-world performance of our method. We use a Flexiv Rizon4 arm with three cameras with different viewpoints as the Figure 10 shows. We finetune the model via LoRA (Hu et al., 2022) with our collected demonstration data following the configuration in the OpenVLA-OFT training process. More configuration and training details are in A.6.5.

**Tasks and Results**   We design four tasks for our evaluation: pick and place, transfer tube, pick up cup and multiple cubes manipulation. Table 2 reports the performance of SpecPrune-VLA in real-world tasks. Our method achieves a 1.70× speedup while maintaining the success rate. The results show that SpecPrune-VLA is a highly potential acceleration method for VLA models.

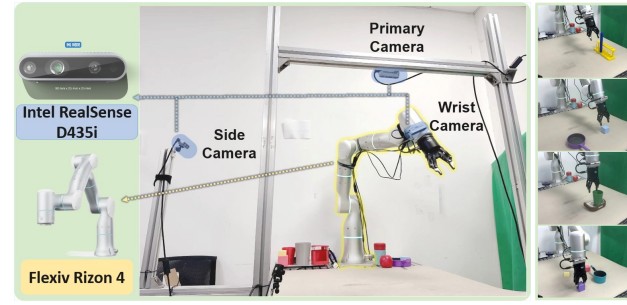

*Figure 10.* Left: Real world robot, a Flexiv Rizon4 arm equipped with a gripper and three Intel cameras. The cameras are mounted separately on the wrist, on its side, and above its head. Right: Four real world evaluation tasks.

## 8. Conclusion

In this paper, we proposed *SpecPrune-VLA*, a training-free acceleration framework that leverages action-aware self-speculative pruning to combine global temporal consistency with local layer-wise importance. Extensive evaluations demonstrate the robust generalizability of our method across *diverse VLA architectures* (OpenVLA-OFT, DB-OFT, CogACT) and *multiple hardware platforms* (NVIDIA A800, RTX 3090/4090). SpecPrune-VLA achieves up to 1.57× speedup in simulation (LIBERO/SimplerEnv) and 1.70× on real-world robotic tasks with negligible success rate degradation, making it a versatile solution for real-time robotic inference.

## Acknowledgments

This work was sponsored by the Shanghai Rising-Star Program (No. 24QB2706200), the National Natural Science Foundation of China (No. 62561160156), and the Shanghai Municipal Commission of Economy and Informatization (No. 2025-GZL-RGZN-BTBX-02035).

## Impact Statement

This paper presents work aimed at advancing the field of Machine Learning. While we acknowledge the potential societal implications of our research, we do not identify any specific negative consequences that require immediate highlighting beyond the general benefits of improved efficiency in Vision-Language-Action models.

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

# A. Appendix

## A.1. Extended Experiment

### A.1.1. EVALUATION ON VARIOUS COMPUTING PLATFORMS

To validate the applicability of our method on different devices, we conduct experiments on NVIDIA GeForce RTX 4090. As illustrated in Table 5, our method achieves an average speedup of **1.48×** in end-to-end latency compared to OpenVLA-OFT. The results consistently demonstrate improved inference efficiency, underscoring the scalability and effectiveness of our approach under diverse computational conditions.

*Table 5.* Performance Evaluation (Success Rate and Speedup) on RTX 4090 GPU

| SimplerEnv | Success Rate (%) | | | | Avg. | Avg. |
|---|---|---|---|---|---|---|
| Method | Spatial | Goal | Object | Long | Speedup | SR(%) |
| OpenVLA-OFT | 92.8% | 97.2% | 97.6% | 95.2% | 1.00× | 95.7% |
| **+ Ours** ($\alpha$=0.8) | 92.8% | 96.6% | 97.2% | 94.4% | 1.48× | 95.3% |

### A.1.2. GENERALIZATION AND ROBUSTNESS ANALYSIS

We conduct experiments on LIBERO-plus (Fei et al., 2025) and VLA-Arena (Zhang et al., 2025a) with prune ratio ($\alpha$) = 0.8. From LIBERO-plus, we choose 200 tasks that add perturbations to camera viewpoint, light condition, background and noise (Figure 11). From VLA-Arena, we choose "Dynamic distractors" and "Dynamic obstacles" level-1 tasks. With degradation in success smaller than 0.6%, the results confirm that our configuration generalizes well to unseen conditions and is robust to perturbations and dynamic scenes.

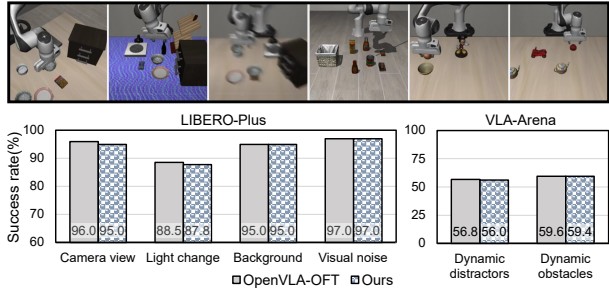

*Figure 11.* Performance evaluation on LIBERO-plus and VLA-Arena benchmark.

### A.1.3. GENERALIZATION OF ACTION MODE RECOGNITION

In Section 6.2, we observe that the execution process can be segmented into fine-grained and coarse-grained modes based on thresholds for translational, vertical, and rotational velocities. Our results in both the **LIBERO simulation** (us-

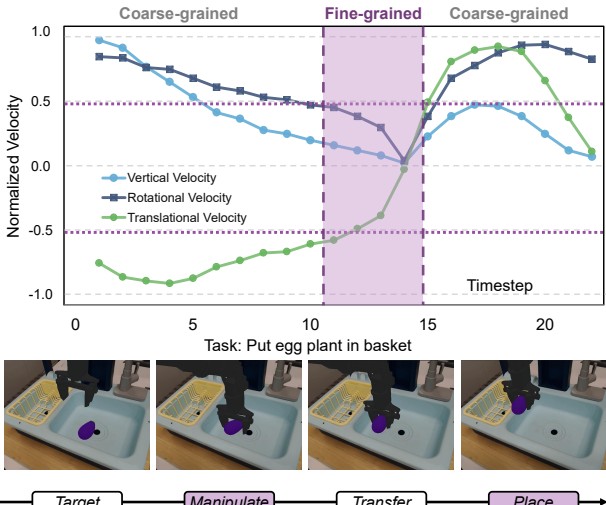

*Figure 12.* The task process is categorized into coarse- and fine-grained actions based on velocity.

ing a **Franka Emika Panda 6-DoF arm**) and **real-world experiments** (using a **Flexiv Rizon 4 7-DoF arm**) demonstrate the generalization of this method. Furthermore, we show that this characteristic is independent of the specific model, robot, or environment. To validate this, we conduct experiments using a different VLA model, UniVLA (Bu et al., 2025) within the **SimplerEnv simulation** framework (Li et al., 2024c). Notably, SimplerEnv employs a **WidowX-250 6-DoF robotic arm**, which differs significantly in kinematics and control characteristics from the previous platforms. As shown in the speed profile in Figure 12, the chosen thresholds consistently distinguish between fine-grained and coarse-grained actions across different setups.

**Velocity threshold distinguishing action mode** Based on empirical data from Figure 12, 6, we set the final thresholds as $v_t^{th} = 0.5$ and $v_r^{th} = 0.2$. When $v_t < v_t^{th}$, $v_r < v_r^{th}$, and $\Delta z \leq 0$, the fine-grained action mode is triggered. An ablation study on the threshold values is conducted on the LIBERO-spatial dataset, as summarized in Table 7.

The results show that when the threshold values are too small, the majority of actions are classified as "coarse-grained," leading to aggressive pruning, which causes a noticeable drop in performance (e.g., success rate drop by 0.6%). Conversely, when the threshold values are too large, nearly the entire process is classified as "fine-grained," resulting in more tokens being retained. Although this yields minimal improvement in success rate, it introduces additional latency. Therefore, the chosen thresholds ($v_t^{th} = 0.5$, $v_r^{th} = 0.2$) achieve an optimal balance between accuracy and efficiency, ensuring high task success while maximizing inference acceleration.

*Table 6.* Performance Evaluation of CogACT (Success Rate and Average Speedup) in the SimplerEnv benchmark, visual matching task

| SimplerEnv Visual Matching | Success Rate (%) | | | | Avg. | Avg. |
|---|---|---|---|---|---|---|
| Method | PickCokeCan | MoveNear | Open/CloseDrawer | AppleDrawer | Speedup | SR(%) |
| CogACT | 91.3% | 85.0% | 71.8% | 50.8% | 1.00× | 74.7% |
| + FastV [ECCV24] | 92.6% | 81.4% | 69.8% | 52.4% | 1.21× | 74.1% |
| + VLA-Cache [NIPS25] | 92.0% | 83.3% | 70.5% | 51.6% | 1.20× | 74.4% |
| **+ Ours** ($\alpha$=0.8) | 91.3% | 83.8% | 71.6% | 50.2% | 1.28× | 74.2% |
| **+ Ours&SC** ($\alpha$=0.8) | 91.7% | 83.8% | 71.2% | 50.0% | 1.42× | 74.1% |

*Table 7.* Ablation study of success rate under different threshold settings on LIBERO-spatial.

| $v_t^{\text{th}}$ | 0.3 | 0.5 | 0.7 |
|---|---|---|---|
| Success Rate | 96.8 % | 97.4 % | 97.6 % |

*(a)* Translational threshold $v_t^{\text{th}}$ (with $v_r^{\text{th}} = 0.2$ fixed).

| $v_r^{\text{th}}$ | 0.1 | 0.2 | 0.4 |
|---|---|---|---|
| Success Rate | 96.8 % | 97.4 % | 97.4 % |

*(b)* Rotational threshold $v_r^{\text{th}}$ (with $v_t^{\text{th}} = 0.5$ fixed).

### A.1.4. EXPERIMENT ON COGACT

In this section, we implement our method on CogACT, a model with another popular architecture. CogACT is built on VLM and has a diffusion action expert. We evaluate on CogACT-Base, a model with Llama2-7b backbone and a 89M diffusion action expert. We use our method in LLM backbone and use static caching (Selvaraju et al., 2024) in action expert. As shown in Table 6, in SimplerEnv, four visual matching tasks, our method alone achieves $1.28\times$ speedup, a little bit lower because the action expert accounts for 25% end-to-end time. While combining with static caching (SC), cache interval = 5 in action expert, our method achieves $1.42\times$ speedup with negligible loss in success rate.

### A.1.5. COMPARISION WITH FLASHATTENTION BASELINE

With `flash-attn==2.5.5` enabled, the wall-clock latency of OpenVLA-OFT on an A800 GPU is 98.5 ms, compared to 109.0 ms for standard attention. Our method achieves a 1.32× speedup over this FlashAttention baseline.

Numerous studies have focused on token pruning or merging in multimodal models, including Vision-Language Models (VLMs) (Liu et al., 2023b), Diffusion Transformers (DiTs) (Peebles & Xie, 2023), and VideoLLMs (Chen et al., 2023). Some approaches, such as FastV (Chen et al., 2024), require accessing attention maps from specific LLM layers. These methods are incompatible with FlashAttention. FlashAttention avoids materializing the full $O(N^2)$ attention matrix in memory by employing tiled computation and online softmax to accumulate results on-the-fly. This design prioritizes memory efficiency and computational speed but does not retain intermediate attention weights. Consequently, recent works such as VisionZip and SparseVLM avoid relying on attention maps for token pruning or merging. This design choice allows them to remain compatible with FlashAttention, thereby enabling further acceleration.

Existing literature primarily targets scenarios involving thousands of visual tokens, such as prefilling with high-resolution images or long videos that result in extensive token sequences. In contrast, current VLA models exhibit different characteristics. The visual input size is typically fixed at 16×16 patches, resulting in 256 tokens per image, as prior studies suggest that higher resolutions do not necessarily yield performance benefits (Kim et al., 2024). During inference, models typically process 1 to 4 images from different viewpoints (e.g., left wrist, right wrist, primary, and side cameras). Even in the case of four images with a sparsity ratio of 30%, the total number of input tokens is approximately 360. In such short sequence length, the GPU compute units (CUDA Cores and Tensor Cores) are not fully utilized, while the additional control flow and non-matrix multiplication operations (such as element-wise ops) introduced by FlashAttention become the dominant factors contributing to latency.

In Figure 13, we profile the latency of both eager attention and flash attention under different input token number and 30% sparsity (same with our method). When the original input length is less than 2048 tokens (corresponding to four images) and a sparsity ratio of 30% is applied on input, eager attention achieves consistently lower latency. This indicates that while visual inputs constitute a significant portion of the current sequence length, they do not yet represent the "long-context" scenarios where FlashAttention provides substantial advantages. Therefore, our sparse pattern delivers significant benefits regardless of the attention mechanism used.

Looking forward, to enable more precise action generation, future models may incorporate additional viewpoints, video sequences, or 3D data. This evolution could increase token counts to 2k, 10k, or even 100k. As illustrated in our results, sparsity methods compatible with FlashAttention will yield increasingly significant acceleration benefits in these long-input regimes.

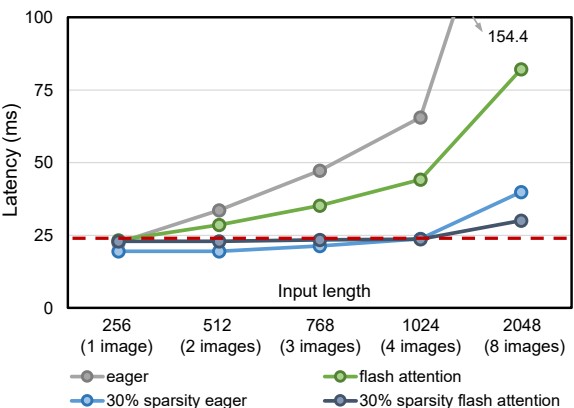

*Figure 13.* The prefilling latency of eager attention and flash attention under different input length on A800 GPU. When original input length is less than 2048 (4 images) and 30% sparsity is applied on input, eager attention achieves consistently lower latency.

### A.2. The overeall algorithm of action-aware pruning

To clearly illustrate the control flow of our action-aware controller and pruning method, we present in Algorithm 1. For each layer of the LLM, the input is the visual tokens from the previous layer (or initial input), and the output is the pruned set of visual tokens.

If the current layer is among the first two layers, we select the top-$K_{local}$ tokens according to their attention scores and add them to the local important token set. After the second layer, we initialize the retained token set $V_{\text{retain}}$ with the top-$K_{global}$ globally important tokens inferred from the previous action step, and then augment it with locally inferred important tokens (top-$K_{\text{base}}$ from Layer 1 and 2) and $K_{dynamic}$ dynamically selected tokens based on low frame similarity. This constitutes the **static pruning at the action level**.

Subsequently, for each layer, we implement **dynamic pruning at layer level**. We dynamically update token importance scores $s_i^{(l)}$ based on layer-wise confidence and intra-layer attention ranking. At designated pruning layers $L_{\text{prune}}$(see A.3), we perform a further pruning step, retaining only the top $\alpha \times \gamma\%$ tokens with the highest cumulative importance scores $S_i^{(l)}$.

The two-level pruning is scheduled by the action-aware controller. It determines the current action mode(fine or coarse grained) and adaptively adjusts the retain token from current local information.

### A.3. Complexity Analysis

Consider a Transformer-based (Vaswani et al., 2017) model with $N$ layers. The computational cost (in FLOPs) per layer when processing $L$ tokens is approximately:

$$\text{FLOPs}_{\text{layer}}(L) = 4LD^2 + 2L^2D + 2LDM + LM$$

where:

- $L$: sequence length (number of tokens)

- $D$: hidden dimension

- $M$: intermediate dimension in FFN

In typical configurations, $M = 3D$ to $4D$ and $D \gg L$ in VLA tasks, making $\text{FLOPs}_{\text{layer}}(L) = 4LD^2 + 2L^2D + 2LDM + LM \approx 10LD^2$ to $12LD^2$. Thus, the computation is primarily driven by the feed-forward and attention projections in the hidden layers, and the overall complexity scales linearly with both sequence length.

---

**Algorithm 1** Action-Aware Pruning in One Layer

**Input**: Full token set $V$ (visual tokens from previous layer or initial input)
**Parameters**: Pruning ratio $\alpha$, Velocity thresholds $v_t^{th}, v_r^{th}$, Decay factor $\beta$, Top-K sizes $K_{local}, K_{dynamic}, K_{global}$ and $\gamma\%$, Layer sets $L_{prune}$
**Output**: Retained token set $V_{retain}$

1: **Action-aware Controller**
2: Compute translational velocity $v_t$ and rotational velocity $v_r$ from action history
3: **if** $v_t < v_t^{th}$ **and** $v_r < v_r^{th}$ **and** $\Delta z \le 0$ **then**
4:    Enter **fine-grained**
5: **else**
6:    Enter **coarse-grained**
7: **end if**
8: **Static Token Pruning at Action Level**
9: $V_{global} \leftarrow$ top-$K_{global}$ tokens based on attention scores from the previous action step
10: Compute frame similarity $Sim(P_t, P_{t-T})$ {Adaptive temporal window based on velocity}
11: $V_{dynamic} \leftarrow$ lowest $K_{dynamic}$ similarity patches
12: For layers $l = 1$ and $l = 2$, compute attention scores:
13:    $V_{local}^{(l)} \leftarrow$ top-$K_{local}$ tokens by attention score in layer $l$
14: $V_{local} \leftarrow V_{local}^{(1)} \cup V_{local}^{(2)}$
15: $V_{retain} \leftarrow V_{global} \cup V_{dynamic} \cup V_{local}$ {Initial retained set}
16: **Dynamic Token Pruning at Layer Level**
17: **if** current layer $l \notin L_{prune}$ **then**
18:    $s_i^{(l)} \leftarrow \omega_{rank,i}^{(l)} \times \omega_{conf}^{(l)}$ {Token importance score}
19:    $S_i^{(l)} \leftarrow (1-\beta) \cdot S_i^{(l-1)} + \beta \cdot s_i^{(l)}$ {Exponential moving average}
20: **else**
21:    $V_{retain} \leftarrow$ tokens with top $\alpha \times \gamma\%$ highest $S_i^{(l)}$
22: **end if**
23: **return** $V_{retain}$

**Static Token Pruning**   The static token pruning strategy reduces an average of 360 tokens from the original about 600 tokens(*i.e.* 60% sparsity in input), let $L_r = 0.4 \cdot L$ denote the number of retained tokens.

*Table 8.* Number of pruned tokens and visual retention rate under different prune ratios.

| Prune Ratio $\alpha$ | 0.6 | 0.8 | 1.0 |
|---|---|---|---|
| Pruned Tokens | 382 | 360 | 336 |
| Visual Retention | 25.4% | 29.8% | 34.4% |

Thus, for layers with $L_r$ length input, the FLOPs become FLOPs($L_r$). For a model with H layers, FLOPs$_{\text{static}} =$ 2FLOPs($L$) + $(H - 2)$FLOPs($L_r$)

**Dynamic Token Pruning**   We apply progressive token pruning in the depth interval $[10, 25]$, with pruning layers selected at regular intervals of $T$. The set of pruning layers is defined as:

$$\mathcal{S} = \{s_k = 10 + (k - 1)T \mid s_k \leq 25, \ k = 1, 2, \ldots\},$$

At each pruning layer $s_k$, we reduce the token count by a retention factor $\gamma = 0.9$. Starting from an initial retained length $L_r$, after the $k$-th pruning step (*i.e.*, at layer $s_k$), the token count becomes $\gamma^k L_r$. The pruning interval is T layers.

The total computational cost is then:

$$
\text{FLOPs}_{\text{final}} = \underbrace{2 \cdot \text{FLOPs}(L)}_{\text{early layers}} + \underbrace{8 \cdot \text{FLOPs}(L_r)}_{\text{shallow layers}}
$$
$$
+ \underbrace{\sum_{k=1}^{|\mathcal{S}|-1} T \cdot \text{FLOPs}(\gamma^k L_r)}_{\text{dynamic pruning}} \quad (12)
$$
$$
+ \underbrace{(H - 10 - (|\mathcal{S}| - 1)T) \cdot \text{FLOPs}(\gamma^{|\mathcal{S}|} L_r)}_{\text{remaining layers}},
$$

where $H$ is the total number of layers, and $L_r$ is the token count after static pruning.

**Overall FLOPs reduction**   In OpenVLA-OFT, the model has 32 layers and T is set to 5. Therefore, FLOPs$_{\text{static}} =$ 0.44 FLOPs($L$), FLOPs$_{\text{final}} =$ 0.85 FLOPs$_{\text{static}} =$ 0.37 FLOPs($L$). Dynamic pruning leads 15% decrease in overall token-level computation across the model. The overall FLOPs reduction in LLM module is 63%.

### A.4. Computation overhead

**Patch similarity:**   Calculating patch similarity introduces extra computations. The cosine similarity is calculated based on the raw patches before the image being encoded.

The similarity between corresponding patches in two frames is computed as $\text{Sim}(P_m^{i,j}, P_n^{i,j}) = \dfrac{P_m^{i,j} \cdot P_n^{i,j}}{\|P_m^{i,j}\|_2 \|P_n^{i,j}\|_2}$ , For N patches and patch size p ,this operation has complexity $\mathcal{O}(N \cdot p^2)$. Here N=256, p=14.

**Attention entropy:**   As defined in Eq. (7), the attention entropy is computed over the $L \times L$ attention matrix at layer $l$. Its complexity is $\mathcal{O}(L^2)$ and introduces 1ms latency. Naively computing it at every step would incur 32ms latency. To avoid this, we observe that entropy patterns are stable across inference steps within the same task in Figure 4I. Therefore, we compute the layer confidence scores only once during the first inference and reuse them in subsequent steps.

Breakdown of every module (Table 9) shows that the computation overhead (calculating patch similarity calculation and attention entropy) only introduces about 4.5% latency on average, which is negligible.

### A.5. Further Analysis on Different View of Images

Different camera viewpoints present distinct characteristics. In this section, we provide a systematic exposition of the first key insight.

**In the fixed view(*e.g.* third-person)**, such as high camera and side camera, the robot arm and the objects it touches are the dynamic components, while the task-related patches (*e.g.* objects on the table) are relatively static, as Figure 14 shows. Therefore, the key is to extract the intersection of both — the regions that involve the dynamic and the task-relevant pixels.

**In the dynamic view(*e.g.* wrist-mounted camera)**, although all objects are in motion, the pixel patches along object boundaries exhibit more significant changes. We process and interpret the temporal signal of a pixel patch using the Fourier transform:

$$P(f) = \int_{-\infty}^{\infty} p(t)e^{-j2\pi ft}dt \quad (13)$$

Here $p(t)$ represents pixel intensity at time $t$ , $P(f)$ denotes the frequency-domain representation of the signal, $f$ stands for frequency. Since objects have different colors from the background, the boundary patches will change color at a certain time. Therefore, in these regions, the resulting signal $p(t)$ contains more abrupt changes, which correspond to higher energy in the high-frequency components of $P(f)$. This indicates that boundary patches carry richer temporal dynamics. As a result, task-related patches and dynamic patches can be complementary to some extent, and their intersection can also be used to identify the most critical regions.

*Table 9.* Latency breakdown per module and comparison with the original model.

| Similarity calculation | First 2 layers | Top-k sorting | Attention Entropy | Our Method | Original Model |
|---|---|---|---|---|---|
| $1.8 \pm 0.1$ ms | $5.5 \pm 0.2$ ms | $2.5 \pm 0.1$ ms | $1.0 \pm 0.2$ ms | 72–78 ms | 109.0 ms |

## A.6. Experiment Details

### A.6.1. IMPLEMENTATION OF COMPARATIVE METHODS

We provide details on the implementation of the baseline and comparative methods. All methods were implemented on one A800 GPU.

**SparseVLM (Zhang et al., 2024b)** SparseVLM is a visual pruning method for Vision-Language models. It dynamically prunes redundant image tokens layer-wise by leveraging self-attention mechanisms and textual guidance.

**FastV (Chen et al., 2024)** FastV prunes visual tokens inVLMs by learning adaptive attention patterns in early layers and removing less attended tokens in subsequent layers, using the LLM's signal to guide pruning in a plug-and-play manner.

**DivPrune (Alvar et al., 2025)** DivPrune formulates token pruning as a Max-Min Diversity Problem (MMDP), selecting a subset of visual tokens that maximizes diversity to reduce redundancy, enabling effective performance at high pruning ratios without fine-tuning or calibration.

**VLA-Cache (Xu et al., 2025c)** While the original method was developed on the OpenVLA model, its authors adapted and extended it to the OpenVLA-OFT. All results reported in our experiments are obtained using the authors' official implementation to ensure reproducibility. It is worth noting that our reported latency and speedup isn't aligned with the data in original paper. This is because the paper uses CUDA Time as metric. According to github issue, CUDA Time only includes the Prefill Stage of the LLM backbone in the model. For example, in auto-regressive model Open-VLA, it is the time of LLM backbone generating the first action token, in OpenVLA-OFT and CogACT, it only includes the LLM backbone during inference. However, we emphasize the end-to-end speedup and report the end-to-end latency, which is calculated starting from the time the model receives the observation to the time the model generate the complete actions that can be executed.

**EfficientVLA (Yang et al., 2025)** The method focuses on VLA models with diffusion action expert and it also optimizes the action expert. Besides, it has not been open-sourced. As a result, we only re-implement it according to the details of visual token pruning and layer pruning provided in the original paper. Following the reported setup,

we retain 28 LLM layers and 112 visual tokens (total of two images) throughout inference.

### A.6.2. PARAMETER SETUP

**Base values for top-Ks** In static token pruning stage, the base value for top-Ks are $K_{global}$=30, $K_{local}$=24 and $K_{dynamic}$=20 according to Section 7.2. The prune ratio $\alpha$ adjust the overall K values by scaling $K_i = \alpha \cdot K_i$. In fine-grained mode, $K_{global}$=30 and $K_{local}$=40 which are higher because the model needs more information to generate fine-grained action. $K_{dynamic}$=10, which is lower because, in fine-grained mode, the speed is much lower, resulting in fewer dynamic tokens will appear.

$$K_{global} = \alpha \times 30$$

$$K_{local} = \alpha \times \begin{cases} 24, & \text{coarse-grained mode,} \\ 40, & \text{fine-grained mode.} \end{cases}$$

$$K_{dynamic} = \alpha \times \begin{cases} 20, & \text{coarse-grained mode,} \\ 10, & \text{fine-grained mode.} \end{cases}$$

**The threshold $\tau$ for Dynamic Token Selection** The threshold $\tau$ for Dynamic Token Selection filters patches with low change magnitude to avoid false positives caused by lighting or camera noise. Since we select the top-$K_{dynamic}$ most dissimilar patches among those below $\tau$, the exact value is not highly sensitive. In simulation, we set $\tau$=0.95 ; in real-world settings with higher noise, we use $\tau$=0.8 . Experiments shows performance remains stable within a range of $\tau \in$[0.9,0.99] in simulation and $\tau \in$[0.6,0.9] in real world, confirming robustness.

**Update Rate $\beta$ in dynamic pruning** Empirically, in exponential moving average (EMA) formulations of the form $S_i^{(l)} = (1 - \beta) \cdot S_i^{(l-1)} + \beta \cdot s_i^{(l)}$, update rate is often set to a small value to ensure smooth calculation. In our method, $\beta$ can be set in a loose range (0.1 - 0.3) without performance loss. However, large value($> 0.6$) will cause loss in task success rate. This is because overly aggressive updates amplify noise from early layers, causing unimportant tokens to be incorrectly assigned high importance scores, while critical tokens may be mistakenly pruned.

The results of different prune ratio are listed in Table 11.

### A.6.3. DETAIL RESULTS

The detailed latency and speedup of baseline and different methods are listed in Table 10.

*Table 10.* Performance Evaluation (Latency and Speedup)

| Method | Latency (ms) / Speedup | | | | Avg. Latency(ms) | Avg. Speedup |
|---|---|---|---|---|---|---|
| | Spatial | Object | Goal | Long | | |
| OpenVLA-OFT | 109.0 / 1.00× | 109.0 / 1.00× | 109.0 / 1.00× | 109.0 / 1.00× | 109.0 | 1.00× |
| SparseVLM | 85.3 / 1.28× | 85.3 / 1.28× | 85.3 / 1.28× | 85.3 / 1.28× | 85.3 | 1.28× |
| VLA-Cache | 101.8 / 1.07× | 101.8 / 1.07× | 101.8 / 1.07× | 101.8 / 1.07× | 101.8 | 1.07× |
| EfficientVLA | 71.7 / 1.52× | 71.7 / 1.52× | 71.7 / 1.52× | 71.7 / 1.52× | 71.7 | 1.52× |
| **Ours($\alpha$= 0.8)** | **74.8 / 1.46×** | **74.5 / 1.46×** | **74.8 / 1.46×** | **75.8 / 1.44×** | **75.1** | **1.46×** |

*Table 11.* Ablation study on prune rate $\alpha$. Success rates (%) and speedup factors are reported per task. Our method achieves optimal trade-off at $\alpha = 0.8$, achieving $1.46\times$ average speedup with minimal accuracy drop.

| Prune Ratio $\alpha$ | Success Rate (%) / Speedup | | | | Avg. SR (%) | Avg. Speedup |
|---|---|---|---|---|---|---|
| | Spatial | Object | Goal | 10 | | |
| None (OpenVLA-OFT) | 97.6 / 1.00× | 96.5 / 1.00× | 97.9 / 1.00× | 94.5 / 1.00× | 96.6 | 1.00× |
| $\alpha$=1.0 | 97.6 / 1.41× | 96.3 / 1.43× | 97.9 / 1.40× | 94.0 / 1.38× | 96.5 | 1.40× |
| $\alpha$=0.8 | **97.6 / 1.46×** | **95.8 / 1.46×** | **97.7 / 1.46 ×** | **93.4 / 1.44×** | **96.1** | **1.46×** |
| $\alpha$=0.6 | 97.4 / 1.52× | 93.2 / 1.50× | 96.0 / 1.52× | 92.2 / 1.51× | 94.7 | 1.51× |

## A.6.4. REAL WORLD EXPERIMENT

**Finetuning** Hyper parameters for OpenVLA-OFT training on real-world tasks are shown in Table 12. We train until the mean L1 loss between predicted and ground-truth normalized actions (scaled between [-1,+1]) falls below 0.01. We decay the learning rate from 5e-4 to 5e-5 after 50K steps.

**Evaluation Details** All evaluations were performed on a single NVIDIA RTX 4090 GPU (24GB VRAM) as the inference machine.

Real-world task details:

1. **"Pick up the cube and place it in the pan."**

   - Task: Use the single-arm robot to grasp the cube and place it in the pan
   - Dataset: 52 demonstrations (52 training)
   - Episode length: 10576 timesteps (353 seconds)

2. **"Remove the cube from the tube rack."**

   - Task: Use the single-arm robot to grasp the tube and carefully place it on the desk
   - Dataset: 50 demonstrations (50 training)
   - Episode length: 10397 timesteps (347 seconds)

3. **"Pick up the purple cube and put it in the pot."**

   - Task: Use the single-arm robot to grasp the purple cube and place it into the pot
   - Dataset: 56 demonstrations (56 training)
   - Episode length: 17129 timesteps (571 seconds)

| Hyperparameter | Value |
|---|---|
| # GPUs | 8 × NVIDIA H200 (141GB VRAM) |
| Learning rate (LR) | 5e-4 (decays to 5e-5 after 50K steps for all tasks) |
| Total batch size | 64 (8 per GPU) |
| # Train steps | "Pick up the cube and place it in the pan.": 80K |
| | "Remove the cube from the tube rack.": 100K |
| | "Pick up the purple cube and put it in the pot.": 80K |
| | "Lift the cup from the tray.": 80K |
| Input images | 1 third-person + 1 side view + 1 wrist camera |
| Image size | 224 × 224 px |
| Obs. history | Single-step (no history) |
| LoRA rank | 32 |
| Action chunk size | 25 steps (open-loop execution) |
| Use proprio | Yes |
| Use FiLM | Yes |
| Trainable params | 853M total: 111M (LoRA) + 269M (action head) + 17M (proprio proj.) + 456M (FiLM proj.) |
| Image augmentations | Random crops (90%), color jitter: |
| | — Brightness: ±0.2 |
| | — Contrast: 0.8–1.2 |
| | — Saturation: 0.8–1.2 |
| | — Hue: ±0.05 |
| | — Crop scale/ratio: [0.9, 0.9] / [1.0, 1.0] |

*Table 12.* OpenVLA-OFT hyperparameters for real-world experiments. Includes parallel decoding, action chunking, continuous actions with L1 regression, and additional inputs (wrist/side cameras + robot state).

4. **"Lift the cup from the tray."**

   - Task: Use the single-arm robot to reach for and lift the cup with a stable grasp
   - Dataset: 51 demonstrations (51 training)
   - Episode length: 11788 timesteps (393 seconds)

Evaluation result: See Figure 15

### A.6.5. EXPERIMENT VISUALIZATION

To understand what the model actually observed when completing a task, we visualize the retained visual tokens. The colored patches are the retained tokens. We show the visualization results of four tasks on four LIBERO datasets in Figure 14. It shows the retained tokens are the task-relevant and dynamic tokens.

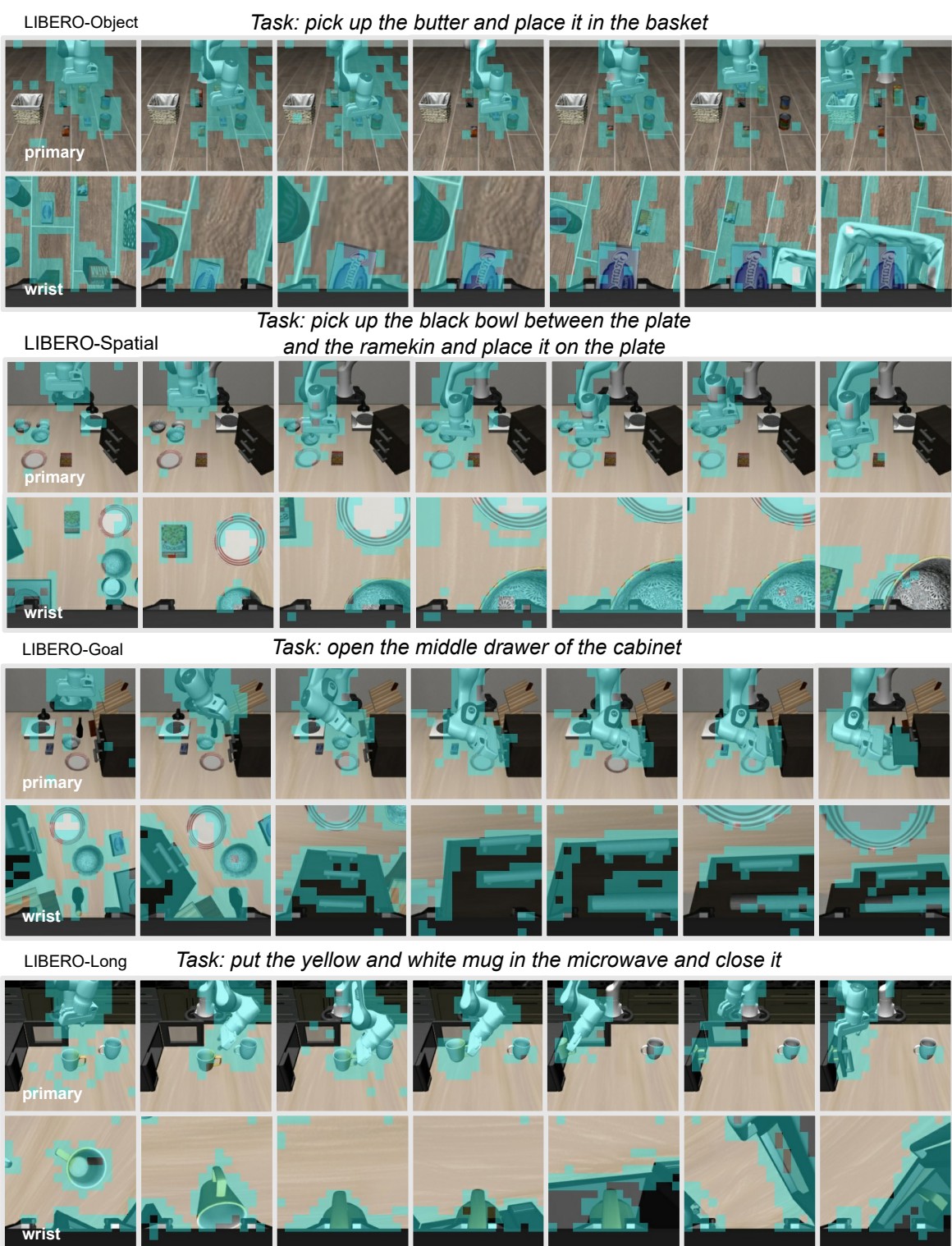

*Figure 14.* Retained tokens during tasks across four datasets and different views.

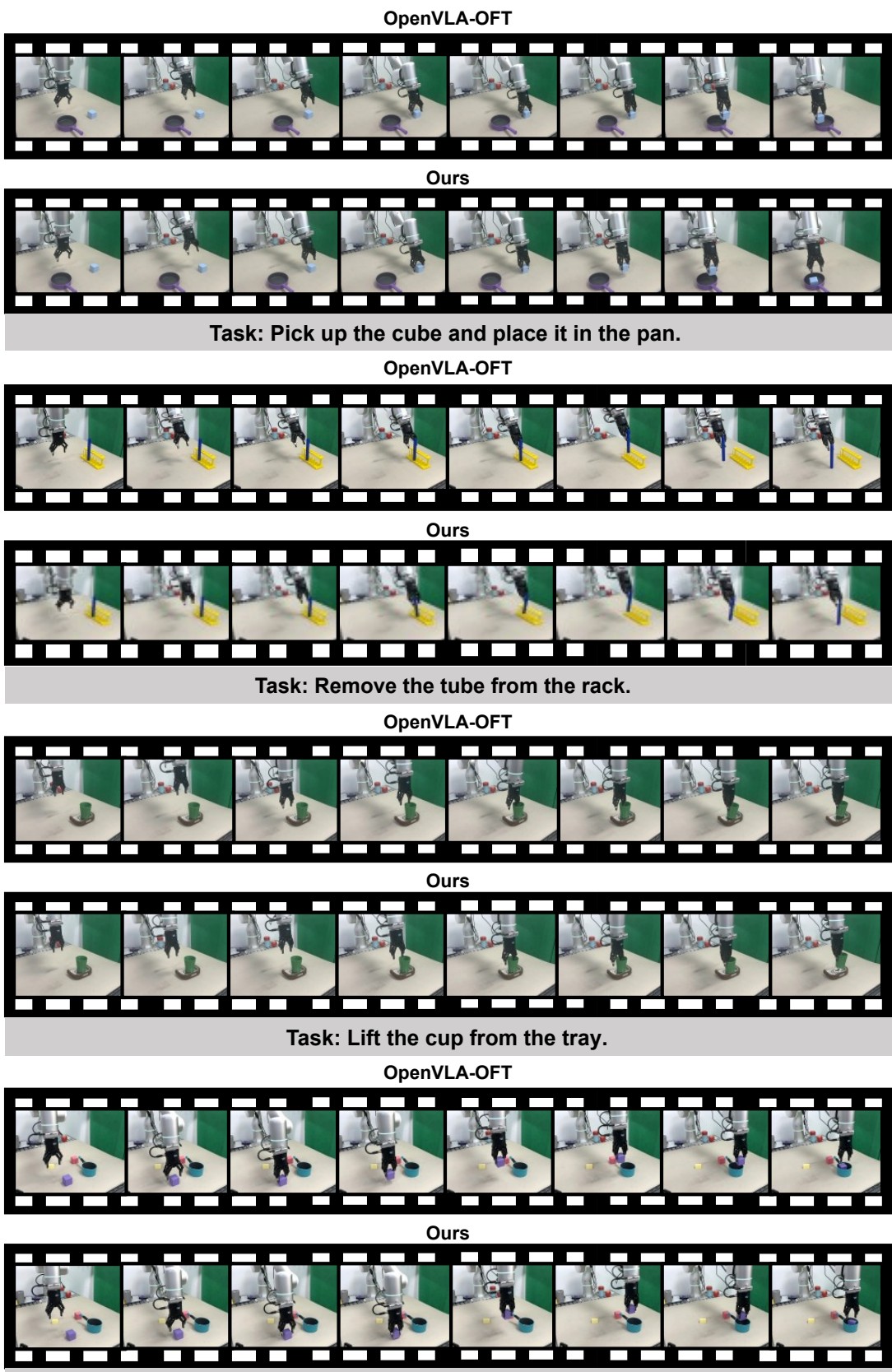

*Figure 15.* The real-world experiment visualization of OpenVLA-OFT and our method across various tasks.

