# OpenReview forum: "SpecPrune-VLA: Accelerating Vision-Language-Action Models via Action-Aware Self-Speculative Pruning"
_ICML.cc/2026/Conference — ICML 2026 regular_

### Official Review · Reviewer_Lv8k · 2026-02-26

**Soundness:** 3
**Presentation:** 4
**Significance:** 3
**Originality:** 3
**Overall Recommendation:** 5
**Confidence:** 4

**Summary:**

This paper tackles the compute-bound inference bottlenecks in recent Vision-Language-Action (VLA) models by proposing SpecPrune-VLA, a training-free, two-stage self-speculative pruning technique. The authors make an interesting observation about spatial-temporal consistency across consecutive steps, leveraging global attention from the previous step to guide action-level static pruning. They pair this with an entropy-based dynamic pruning approach at the layer level, as well as a lightweight controller that adaptively adjusts the pruning aggressiveness based on the robot's end-effector speed. Overall, the experiments show that the method significantly accelerates inference in both simulation and real-world robotic tasks, all without suffering any meaningful drop in task success rates.

**Compliance With Llm Reviewing Policy:**

Affirmed.

**Ethical Review Flag:**

Flag this paper for an ethics review.

**Final Justification:**

The authors' responses have sufficiently clarified my doubts, ensuring the retention of my positive score.

**Key Questions For Authors:**

1. What are the specific structural reasons for the lower acceleration efficiency observed in $\pi_0$ compared to OpenVLA-OFT?
2. Would the proposed velocity-based controller thresholds remain effective and generalizable in extremely dynamic scenarios, such as throwing and catching objects or performing high-speed bimanual assembly?

**Limitations:**

The manuscript lacks a dedicated limitations section and would be much stronger if it explicitly addressed failure modes like sudden camera occlusions.

**Strengths And Weaknesses:**

Strengths:
1. It solves the information loss issue seen in previous research methods that relied heavily on single-frame shallow layers. Reusing globally important tokens from previous steps captures the continuous nature of VLA tasks.
2. The method smartly combines information theory and robot kinematics. It uses attention entropy to reliably filter noise across layers, while a highly efficient, velocity-based controller dynamically preserves visual fidelity during critical fine-grained actions without complex overhead.
3. This approach successfully breaks the notorious speed-accuracy trade-off. It demonstrates impressive robustness and acceleration across both the benchmarks and real-world setups.

Weaknesses:
1. Eq. 3 relies on pixel-level cosine similarity to extract dynamic tokens. The authors try to patch its well-known vulnerability to camera noise and lighting shifts using a velocity-based sampling heuristic and a threshold. However, pixel-wise comparison is fundamentally brittle. I remain questioning that these workarounds can reliably prevent the model from misclassifying random noise as dynamic tokens in unstructured, real-world deployments.
2. While the authors assert that the method is platform-independent due to normalized velocity data, the thresholds governing coarse and fine control are determined strictly empirically. There is a strong likelihood that this assumption will break down in robotic tasks that demand rapid and highly dynamic movements.
3. The real-world evaluation is confined to four relatively static and simplistic tabletop manipulation tasks, which limits the demonstration of the method's robustness in more complex physical environments.

---

> ### Author Rebuttal · Authors · 2026-03-29
>
> ## Re to W1:
> We fully acknowledge the concern regarding the unpredictable light changing and random noise in real-world deployment. Therefore, we employ velocity-based frame sampling to augment the real changing part in image. To simulate the real world situation, We also evaluate on benchmark that adds perturbations to Light and Noise(Section 7.6) and see limited drop (0.6%) in performance. Our real-world experiment also indicates the robustness of our method. The average success rate drop only by 1%.
> Combined with Weakness 3, we add a more difficult tasks in **Re to W3** part. The result shows that our method effectively mitigates the impact of random noise that would otherwise lead to task failure.
>
> While we acknowledge that extreme noise could theoretically lead to misclassification, our combination of adaptive temporal sampling and similarity thresholding (τ) limits this probability to a negligible level, as evidenced by our stable performance across diverse settings.
>
> ## Re to W2 and Q2:
> We fully understand your concern under dynamic scene. When the model manipulation object is dynamic, the model's end effector may move with it and thus remain at a high speed. In such cases, (1) the dynamic object can be largely captured by our dynamic detection module. The targeting process is categorized as coarse-grained, which is aligned with our idea. (2) When the end effector is touching the object, the velocity is relative lower and would be categorized as fine-grained action.
>
> Furthermore, we evaluate on two dynamic scene in VLA-Arena[1], the result shows that under normal dynamic scene, our method won't miss the target and can ensure precise manipulation.
> |VLA-arena|Success Rate||SpeedUp|
> |-|:-:|:-:|:-:|
> ||Dynamic Distractors|Dynamic obstacles|Average
> |OFT|56.8%|59.6%|1.00 $\times$|
> |ours|56.0%|59.4%|1.40 $\times$|
>
> While extremely dynamic scenarios(e.g., catching, ball-playing) remain an open challenge for most existing Vision-Language-Action (VLA) models, we agree that this constitutes a specific limitation of our current approach. We are considering improving the action mode detction by relative speed calculation or integrating lightweight learning-based mode classifiers and will **explicitly add a discussion on this failure mode in the 'Limitations'**
>
> ## Re to W3:
> These tasks (pick&place and transfer) are widely used ([2][3]) for VLA testing. For further evaluation, we add a more difficult real-world task: In the same environment with TransferCube task in Figure 14, we let the model to pick up the test tube from the rack and place it into the slot. It needs more precise control to position into the hole on the slot. The result of success rate is as follows: **OpenVLA-OFT: 74.8%, Ours: 74.0%**.
>
> ## Re to Q1:
> Thank you for the opportunity to further clarify our results. $\pi_0$ has a flow matching expert and needs iterative denoising. The action expert shares kvcache with the LLM backnone. In LLM backbone, pruning leads to linear reductions in QKV ( $O(L)→O(L_{pruned​})$ ) and FFN( $O(L)→O(L_{pruned}​) $ ), plus a quadratic reduction in self-attention ( $O(L^2)→O(L^2_{pruned}​) $. In action expert the reduction derives from the linear computation reduction in attention matrix, from $O(A \cdot L)$ to $O(A \cdot L_{pruned})$ (A denotes the number of action tokens).
>
> For models with flow-matching or diffusion action expert, we can adopt distillation or caching (widely used) to accelerate the overall inference. For example, we evaluate on CogACT-Base[4], a model with Llama2-7b backbone and a 89M diffusion action expert. We use our method in LLM backbone and use uniform caching (Interval=5) in action expert. In SimplerEnv, four visual matching tasks, our method alone achieves 1.30 $\times$ speedup, a little bit lower because the visual input length is shorter and the action expert accounts for extra 25% time. While combining with feature caching in action expert, our method achieves 1.42 $\times$ speedup with negaligible loss in success rate.
>
> |SimplerEnv|||**success rate**|||\||SpeedUp|
> |:-:|:-:|:-:|:-:|:-:|:-:|:-:|:-:|
> |Method|PickCokeCan|MoveNear|Open/CloseDrawer|DawerApple|Average|\||Average|
> |CogACT|91.3%|85.0%|71.8%|50.8%|74.7%|\||1.00 $\times$|
> |Ours|91.3%|83.8%|71.6%|50.6%|74.3%|\||1.30 $\times$|
> |Ours+cache|91.7%|83.8%|71.2%|50.0%|74.1%|\||1.42 $\times$|
>
> [1]Zhang, Borong et al. “VLA-Arena: An Open-Source Framework for Benchmarking Vision-Language-Action Models.” ArXiv abs/2512.22539 (2025): n. pag.
>
> [2]Xu S, Wang Y, Xia C, et al. Vla-cache: Towards efficient vision-language-action model via adaptive token caching in robotic manipulation[J]. arXiv e-prints, 2025: arXiv: 2502.02175.
>
> [3]Kim M J, Pertsch K, Karamcheti S, et al. Openvla: An open-source vision-language-action model[J]. arXiv preprint arXiv:2406.09246, 2024.
>
> [4]Li, Qixiu et al. “CogACT: A Foundational Vision-Language-Action Model for Synergizing Cognition and Action in Robotic Manipulation.” ArXiv abs/2411.19650 (2024): n. pag.

---

> > ### Author Rebuttal · Reviewer_Lv8k · 2026-04-02
> >
> > Thanks for addressing my concerns in the rebuttal; I'll maintain my original positive score.

---

> > > ### Author Response · Authors · 2026-04-02
> > >
> > > We thank the reviewer for the positive acknowledgmen. We are glad that our rebuttal has fully addressed the concerns.

---

### Official Review · Reviewer_5CZF · 2026-03-03

**Soundness:** 1
**Presentation:** 1
**Significance:** 1
**Originality:** 1
**Overall Recommendation:** 2
**Confidence:** 5

**Summary:**

This paper attempts to address an important and practical problem in Vision-Language-Action (VLA) models: inference acceleration. The authors observe that the LLM backbone is the primary compute bottleneck and propose SpecPrune-VLA, a training-free token pruning pipeline.

The proposed method consists of three main components: (1) Action-level Static Pruning. At the beginning of each inference step, the model selects an initial subset of visual tokens by combining three sources of information: globally important tokens, dynamic tokens identified via inter-frame patch similarity, and locally important tokens guided by the current step's shallow-layer attention. (2) Layer-level Dynamic Pruning. As the remaining tokens propagate through the LLM, the model prunes them based on a dynamically updated importance score, which is calculated using intra-layer attention ranking and layer-wise confidence (attention entropy). (3) Action-aware Controller. A rule-based controller regulates the pruning aggressiveness based on the end-effector's velocity.

The authors evaluate SpecPrune-VLA on the LIBERO simulation benchmark and a real-world robotic setup, demonstrating empirical speedups with minimal impact on the task success rate.

**Compliance With Llm Reviewing Policy:**

Affirmed.

**Ethical Review Flag:**

Flag this paper for an ethics review.

**Final Justification:**

The rebuttal does not resolve our main concerns about method novelty, parameter sensitivity, and comparison fairness. The requested same-parameter transfer experiments are still not adequately addressed. Moreover, the claimed torch.compile compatibility remains insufficiently substantiated, especially without concrete implementation evidence, making the reported results difficult to verify. Finally, the amount of new results added in the rebuttal suggests that substantial revision of the paper would be necessary. Thus, I maintain reject.

**Key Questions For Authors:**

1. Considering the massive hyperparameters introduced, how do the exact same parameters transfer to other model-benchmark combinations (e.g., standard OpenVLA on LIBERO/SimplerEnv, or OpenVLA-OFT on SimplerEnv/Meta-World)? Additionally, since VLA performance varies across hardware, please test these exact parameters on other devices (e.g., RTX 5090 or A100).
2. Why is there no empirical comparison with SP-VLA?
3. When evaluating the π0 model, why is there no comparison with other acceleration baselines?
4. The efficient inference of π0 relies on torch.compile, which is incompatible with dynamic token pruning. How did you resolve this fundamental incompatibility? Was torch.compile disabled during your evaluation, and if so, is the baseline comparison still fair?
5. Since the proposed method requires the full attention matrix, it is incompatible with FlashAttention. How does the actual wall-clock latency of your method compare to a standard VLA baseline equipped with FlashAttention?
6. Does the reported inference speedup account for the computational overhead introduced by the heuristic modules (e.g., computing patch-level cosine similarity, sorting attention scores, and calculating layer-level entropy)? Please provide a more detailed latency breakdown.


Note: If the authors can provide the requested experiments, I may consider raising my score to 3. However, in my view, the proposed method suffers from fundamental limitations.

**Limitations:**

Overall, the proposed method is a combination of existing techniques, offering marginal technical contribution and few meaningful insights to advance the field. Consequently, this complex combination inevitably introduces an excessive number of sensitive hyperparameters, limiting its practical value.  The lack of transfer experiments further raises concerns. While I acknowledge the authors' efforts in the experiments and figures, my overall recommendation is 2 (Reject).

**Strengths And Weaknesses:**

**Strengths**
1. The paper attempts to tackle a practical problem: inference acceleration for VLA models, with evaluations on both simulation and real robots.
2. The authors put effort into the figures to illustrate their method.




**Weaknesses**
1. The proposed method is a heuristic-driven combination of existing techniques (attention entropy, transformer layer filtering, frame differencing, token importance score calculation,...). Rather than offering a technical  breakthrough, the paper reads more like a technical report on testing and tuning existing model acceleration tricks for VLA.  Thus, it hardly constitutes a technical contribution.
2.  As a direct consequence of this heuristic approach (Weakness 1), the method introduces an excessive number of sensitive hyperparameters (K, α, β, τ, γ, and v).  Because robotics environments are dynamic and diverse, a rigid set of parameters fails to adapt across different tasks or even across different timesteps within a single episode.  Therefore, this heavy reliance on extensive manual tuning renders the method highly impractical and severely limits its generalizability.
3. The experiments are primarily conducted using OpenVLA-OFT on LIBERO. Given the method's heavy reliance on sensitive hyperparameters (Weakness 2), it is highly likely that this specific configuration is overfitted and this exact parameter set cannot maintain performance when transferred to other setups (e.g., OpenVLA on LIBERO, or OpenVLA-OFT on SimplerEnv).
4. The writing of this paper needs much improvement as it introduces forced jargon that hinders reading. To name a few, simple adjacent frame similarity is unnecessarily inflated to "spatial-temporal consistency"; the motivation in the abstract regarding "local information" and "global context" is highly ambiguous; the core concept of "two-level pruning" is poorly articulated, and so on. These writing flaws create significant barriers to reading and comprehension.
5. No code is provided, raising severe reproducibility concerns, especially given the massive hyperparameters introduced.
6. This method does not support FlashAttention.

---

> ### Author Rebuttal · Authors · 2026-03-29
>
> ## Re to W1:
> The core novelty lies in our adaptive, fine-grained token pruning strategy, which is fundamentally guided by two key insights specific to VLA models: Spatial-temporal consistency and hierarchical attention patterns.
> These insights directly dictate our token importance scoring mechanism. Specifically, the scoring is not a generic application of heuristics but is tailored to capture redundancy across frames and layers based on our observations. Regarding the implementation details, standing on the shoulders of giants, we acknowledge utilizing existing concepts, such as information entropy H(⋅) and Exponential Moving Average (EMA). It allows us to build a more stable and efficient framework. Our technical contribution is deriving a unified pruning framework grounded in our insights, addressing the unique challenges of VLA acceleration.
>
> ## Response to Q1, W2 and W3:
> We admit that the hyperparameters exist but it highlights the physical meaning of our method. Their selection is driven by data-driven physical insights and quantifiable metrics rather than arbitrary empirical tuning. The main parameters top-Ks are selected based on the goal of maximizing the Recall of important tokens(metioned in Sec.3.2) between consecutive inferences; prune ratio α trades off success rate vs. speedup; The velosity thresholds represent the physical movement pattern, rather than purely tuning.
>
> In our paper, we evaluate on LIBERO and LIBERO-plus. Additionally, we validate generalization with identical parameters on CogACT-SimplerEnv and OpenVLA-OFT-SimplerEnv combinations.
> |SimplerEnv|||**success rate(%)**|||\||SpeedUp|
> |:-:|:-:|:-:|:-:|:-:|:-:|:-:|:-:|
> |Method|PutCarrot|PutSpoon|StackCube|PutEggplant|Average|\||Average |
> |OpenVLA-OFT|64.1|89.2|30.8|97.5|70.4|\||1.00 $\times$ |
> |Ours|63.8|88.7|30.0|98.0|70.1|\||1.44 $\times$|
>
> |SimplerEnv|||**success rate(%)**|||\||SpeedUp|
> |:-:|:-:|:-:|:-:|:-:|:-:|:-:|:-:|
> |Method|PickCokeCan|MoveNear|Open/CloseDrawer|DawerApple|Average|\||Average|
> |CogACT|91.3|85.0|71.8|50.8|74.7|\||1.00 $\times$|
> |Ours|91.3|83.8|71.6|50.6|74.3|\||1.30 $\times$|
>
> Hardware generalization (RTX 4090, same parameters):
>
> |LIBERO|||**success rate(%)**|||\||SpeedUp|
> |:-:|:-:|:-:|:-:|:-:|:-:|:-:|:-:|
> |Method|Spatial|Goal|Object|Long|Average|\||Average|
> |OpenVLA-OFT|92.8|97.2|97.6|95.2|95.7|\||1.00 $\times$|
> |Ours|92.8|96.6|97.2|94.4|95.3|\||1.48 $\times$|
>
> ## Re to Q2:
> SP-VLA was proprietary during our experiments but has since been open-sourced. We integrated it into OpenVLA-OFT, yielding the following results:
> |SP-VLA|Spatial|Object|Goal|Long|
> |:-:|:-:|:-:|:-:|:-:|
> |SR(%)|84.2|78.0|81.6|66.6|
> |latency(ms)|82.8|83.0|85.2|78.4|
> |speedup($\times$)|1.32|1.31|1.28|1.39|
>
> SP-VLA causes significant SR drop because its lightweight action generator introduces errors in chunked-action models.
> ## Re to Q3:
> $\pi_0$'s codebase has a completely different style with other open-source VLA model. Therefore, due to the time limit, we didn't implement other methods. Now we provide result of DivPrune and VLA-Cache for comparision. The result is consistant with our evaluation in Sec.7.4.
> |LIBERO|Spatial|Object|Goal|Long|Avg SR(%)|Avg SpeedUp($\times$)|
> |:-:|:-:|:-:|:-:|:-:|:-:|:-:|
> |$\pi_0$|96.8|98.8|95.8|85.2|94.2|1.00|
> |DivPrune|91.4|92.8|89.0|79.6|88.2|1.33|
> |VLA-cache|96.5|96.6|96.6|84.6|93.6|1.21|
> |Ours|96.6|98.0|95.2|84.2|93.5|1.31 |
> ## Re to Q4:
> To accommodate torch.compile's fixed-shape requirement, we empirically observe per-layer pruned token counts and set fixed sequence lengths per layer. This adaptation reduces SR by only 0.2% while ensuring fair comparison. We will clarify this in the paper.
> ## Re to Q5:
> With flash-attn==2.5.5 enabled, OpenVLA-OFT's wall-clock latency is 98.5ms on A800 (vs. 107.0ms standard attention). Limited speedup stems from relatively short input sequences vs. video tasks (tens thousands of tokens). Our method achieves 1.32$\times$ speedup over the FlashAttention baseline.
> ## Re to Q6:
> Yes, as mentioned in 7.4, the speedup is reported as end-to-end latency including overhead. The overhead is reported in A.3 COMPUTATION OVERHEAD. We list below:
> |Similarity calculation|Top-k sorting|Attention Entropy|Ours latency|Original latency|
> |:-:|:-:|:-:|:-:|:-:|
> |$1.8\pm0.1\text{ms}$|$2.5\pm0.1\text{ms}$|$1.0\pm0.2\text{ms}$|$72\text{--}78\text{ms}$|$109.0\text{ms}$|
>
> The data shows the computation overhead only introduces about 6.8% latency on average, which is negligible.
> ## Re to W4:
> We appreciate the reviewer's sharp eye on the writing flaws. We would like to clarify that spatial-temporal consistency isn't simply the adjaacent frame similarity, it also points to our observation: similiar attention pattern of the whole model between inference. For other flaws we will correcte the overstated terminology and clarify the definition.
> ## Re to W5:
> We guarantee the authenticity and reproducibility of our results. We commit to making the source code publicly available upon the acceptance of this paper.

---

> > ### Author Rebuttal · Reviewer_5CZF · 2026-04-02
> >
> > I appreciate the additional experiments and clarifications. However, my concerns are only partially resolved, and several key issues remain unclear.
> >
> > First, the rebuttal still does not sufficiently address my main originality concern. The method continues to appear as a heuristic combination of existing ingredients, and the response does not clearly explain what the fundamental technical contribution is beyond integrating these components for VLA acceleration.
> >
> > Second, the response on hyperparameters and generalization is still incomplete. My concern was not only that some parameters have intuitive meanings, but that the overall method introduces many sensitive design choices and thresholds. The rebuttal mainly discusses top-K values, pruning ratio, and velocity thresholds, but does not sufficiently address the robustness of the full parameter set or explain how these parameters were fixed before evaluation. In addition, the new transfer results are only partial relative to the settings I asked about, so I still do not have strong evidence that the same configuration generalizes broadly across models, benchmarks, and hardware.
> >
> > Third, I find the added SP-VLA comparison unclear. The explanation that SP-VLA causes errors because of its “lightweight action generator” is confusing to me, and I am not sure this is the correct interpretation of the baseline difference. Please clarify the exact implementation and evaluation protocol used for SP-VLA, and whether the comparison is fully fair.
> >
> > Fourth, the additional comparison on π0 is helpful, but it still leaves questions. It is not fully clear which baselines were implemented under exactly the same setting and whether the comparison protocol is identical. Moreover, the new results seem to show a tradeoff rather than a clear advantage, so I would like the authors to clarify what the main claim is in the π0 setting: higher speed, better accuracy-speed balance, or stronger robustness.
> >
> > Fifth, the torch.compile issue is still not fully resolved for me. The rebuttal says fixed sequence lengths were set empirically per layer, but it remains unclear whether torch.compile was enabled for both the baseline and the proposed method, how these fixed lengths were determined, and whether this adaptation changes the nature of the dynamic pruning method itself.
> >
> > Sixth, the FlashAttention response is also incomplete. The rebuttal reports the FlashAttention baseline latency and a relative speedup over that baseline, but does not provide the absolute latency of the proposed method under the same setting.
> >
> > Overall, while the additional experiments are useful, I do not think the rebuttal fully resolves my concerns about originality, parameter sensitivity, fairness of comparison, and implementation compatibility.

---

> > > ### Author Response · Authors · 2026-04-02
> > >
> > > ### First
> > > We clarify that our method is a principled framework driven by VLA-specific insights, not a heuristic combination:
> > >
> > > From Local to Global Semantics: Unlike methods (e.g., FastV) relying on local shallow features that often miss critical objects (Sec 3.1), our Action-Level Static Pruning leverages Spatial-Temporal Consistency to preserve global(whole model), task-relevant semantics.
> > >
> > > Entropy-Weighted Scoring: Combining with our observation of Hierarchical Attention Patterns, we propose an Entropy-Weighted Mechanism that adaptively weighs tokens based on attention confidence, capturing the shift from shallow visual features to deep action semantics.
> > >
> > > Velocity-Based Frame Sampling: Unlike naive frame differencing, which is sensitive to sensor noise and lighting variations, our Velocity-Guided strategy directly links the perception rate to the intensity changes across frames. This enables us to avoid erroneously retaining interference tokens while accurately identifying genuine dynamic tokens.
> > >
> > > To our knowledge, we are the first to approach VLA pruning from a global perspective. Ablation studies (Appx A.1) on Global Attention Reuse and Entropy-based weighting confirm these are not heuristic additions but essential components for maintaining VLA performance.
> > > ### Second
> > > We address the two remaining concerns below:
> > >
> > > **The threshold $\tau$ for Dynamic Token Selection:** Filters low‑change patches to avoid noise. Since we select the top-$K_{dynamic}$ most dissimilar patches among those below $\tau$ , the exact value is not sensitive. Experiments shows performance remains stable within a range of $\tau \in$[0.9,0.99] in simulation and $\tau \in$[0.6,0.9] in real world, confirming robustness.
> > >
> > > **Update Rate $\beta$ in dynamic pruning:** Empirically, in exponential moving average (EMA), update rate is often set to a small value to ensure smooth calculation. In our method, $\beta$ works well in a loose range (0.1 - 0.3). When $\beta$>0.6 overly aggressive updates amplify early-layer noise, causing unimportant tokens to receive high importance scores.
> > >
> > > **Dynamic prune rate $\gamma$**: We initially set $\gamma$ to a relative large value 0.9. And we ablate it with overall prune rate $\alpha$(Sec.7.3) by scaling $\gamma$($\alpha$)=$\alpha \cdot \gamma$.
> > >
> > > **Robustness of our method**: In Fig. 7, Sec.7.5, we show that under different prune rate, our method consistently maintain a high succcess rate across various tasks.
> > >
> > > We evaluate consistently across three models ($\pi_0$, OpenVLA‑OFT, CogACT), three benchmarks (LIBERO, SimplerEnv, real world), and three GPUs (A800, RTX 3090, RTX 4090). We believe this constitutes strong evidence that SpecPrune‑VLA generalizes broadly.
> > > ### Third
> > > In the  SP-VLA's failure cases, we consistently observed that the robot arm's movement was often "too ahead" of the target object during grasping attempts. Therefore, we traced the failures to its lightweight action generator (using Ridge Regression to produce approximate linear actions), which caused overshot and targetting error.  Implementation detail is in reply to Reviewer jTvd's Ack. We ensure we follow the source code and the comparison is fully fair.
> > > ### Fourth
> > > We respectfully clarify that all baselines were evaluated under fair protocols: As VLA-Cache is originally developed for VLA models, we used its original parameters. DivPrune is proposed for VLM models, therefore we set it to the same 70% image token sparsity as our method.
> > >
> > > **Our claim is better accuracy-speed balance**, demonstrated through two controlled comparisons:
> > > * Against VLA-Cache: At comparable success rates (93.5% vs 93.6%), our method achieves higher speedup (1.31× vs 1.21×).
> > > * Against DivPrune: At comparable sparsity (70%) and speedup (1.31× vs 1.33×), our method achieves higher success rate (93.5% vs 88.2%).
> > > ### Fifth
> > > We clarify below:
> > >
> > > (1) torch.compile was enabled for both baseline and our method. (2) Fixed sequence lengths are empirically derived—under α=0.8, coarse/fine modes retain 186/288 tokens on average after layer 2, set as fixed lengths. In the following dynamic pruning stage, the dynamic pruning rate γ is deterministic, subsequent layer input lengths are fully determined. (3) This adaptation does not change the nature of our method. Our core idea: progressively pruning to action-centric tokens based on attention ranking and layer confidence, remains unchanged. It causes only a 0.2% drop in success rate, confirming negligible impact.
> > > ### Sixth
> > > Under the same settings in the paper, we already provide the latency(ms) of our method in Tab.10, A.5.3 in our paper. We list here: Spatial:74.8, Object:74.5, Goal:74.8, Long:75.8, Average:75.1 .
> > > ### Finally
> > > We respectfully note that we have already provided the actual numerical latency breakdown in our first rebuttal in the table right below the "we list below".
> > > Similarity calculation: 1.8±0.1 ms, top‑k sorting: 2.5±0.1 ms, attention entropy:1.0±0.2 ms. Our method: 72–78 ms and original latency: 109.0 ms.

---

### Official Review · Reviewer_jTvd · 2026-03-08

**Soundness:** 2
**Presentation:** 2
**Significance:** 2
**Originality:** 2
**Overall Recommendation:** 3
**Confidence:** 4

**Summary:**

This work proposes a training-free two-level token pruning framework for Vision-Language-Action (VLA) model acceleration, addressing the critical compute bottleneck in LLM backbones of VLA models. The method leverages spatial-temporal consistency and action granularity to balance speedup and task success rate, with promising empirical results on simulation and real-world robotic tasks.

**Compliance With Llm Reviewing Policy:**

Affirmed.

**Final Justification:**

I share similar concerns with Reviewer 5CZF regarding the novelty of the proposed method, parameter sensitivity analysis, and fairness of experimental comparisons. These issues remain inadequately addressed in the current revision. Therefore, I maintain my original rating.

**Key Questions For Authors:**

See the weaknesses

**Limitations:**

See the weaknesses

**Strengths And Weaknesses:**

Strengths:
(1) This paper identifies the LLM backbone as the dominant inference latency source (70%) for VLA models, which is important for realtime robot tasks.
(2) The proposed method divides tokens into static and dynamic tokens to prun tokens at different stages, which is reasonable.

Weakness:
(1) This paper relies heavily on empirically chosen heuristics and hyperparameters with minimal theoretical or principled justification, a critical limitation for ICML a top-tier ML conference. For example, the selection of 15th and 32nd layers for global attention reuse is arbitrary and no analysis is provided for why these layers capture the most task-relevant and action-centric tokens. The velocity-based frame sampling formula and action granularity thresholds are purely empirical.
(2) The proposed method only use the end-effector velocity the to identify the fine-grained stage. However, it is not sutiable for dynamic scenes with dynamic objects, which is important for VLA models on processing dynamic tasks.
(3) More simulation benchmarks should be validated. The single LIBERO benchmark can not fully demonstrate the effectiveness of VLA models, this paper should provide more experiments on other benchmarks (e.g. CALVIN)
(4) This paper only compares OpenVLA-OFT on the real-world development. The proposed method should be compared with  more methods (e.g., VLA-Cache, RTC).

---

> ### Author Rebuttal · Authors · 2026-03-29
>
> ## Re to W1:
> We appreciate the reviewer's insightful comment regarding the hyperparameters. We clarify that our parameters are not arbitrarily tuned but are grounded in data-driven physical insights, quantifiable metrics, and principled design choices specific to robotic manipulation.
>
> Specifically:
> * The main parameters top-Ks are selected based on the goal of maximizing the Recall of Important Tokens, a quantifiable metric measuring the spatial-temporal overlap between consecutive steps (Section 7.2). The prune ratio indicates how much part of image should be retained, serving as the trade-off between success rate and speedup.
> * For the selection of layers, we point out in our observation that there is hierachical attention pattern in vla models, the shallow layers attend more on background, middle layers focus on task relevant part and deep layers focus on action centric objects. The specific layers has minimal influence.
> * The Action Granularity: These thresholds distinguish between kinematic phases: coarse-grained free-space motion (high velocity) and fine-grained contact manipulation (low velocity). As shown in Figure 6 adn 12, robot trajectories naturally exhibit a bimodal velocity distribution. Thresholds are set at the valley of this distribution, representing a physical boundary in movement patterns rather than a mathematical hyperparameter.
> * The velocity-based frame sampling formula is a principled first-order approximation of the inverse relationship between robot speed and valid temporal context. The linear form efficiently captures this adaptive temporal consistency, with constants derived from typical manipulator control frequencies, ensuring the reference frame always falls within a physically meaningful window without requiring dataset-specific tuning.
>
> Also, we provide the experiment on a new model CogACT in a different simulation benchmark SimplerEnv to show the generalization of our methods and setings.
>
> |SimplerEnv|||**success rate**|||\||SpeedUp|
> |:-:|:-:|:-:|:-:|:-:|:-:|:-:|:-:|
> |Method|PickCokeCan|MoveNear|Open/CloseDrawer|DawerApple|Average|\||Average|
> |CogACT|91.3%|85.0%|71.8%|50.8%|74.7%|\||1.00 $\times$|
> |Ours|91.3%|83.8%|71.6%|50.6%|74.3%|\||1.30 $\times$|
> |Ours+Cache|91.7%|83.8%|71.2%|50.0%|74.1%|\||1.42 $\times$|
>
> In SimplerEnv, four visual matching tasks, our method alone achieves 1.30 $\times$ speedup. A little lower because the diffusion action expert accounts for another 25% time. Since our method is orthogonal to accelerartion technique in diffusion model such as feature caching, when combining with uniform feature caching (interval=5), we achieve 1.42 $\times$ speedup with negaligible loss in success rate.
>
> ## Re to W2:
> We fully understand your concern under dynamic scene. When the model manipulation objecy is dynamic, the model's end effector may move with it and thus remain at a high speed. In such cases, (1) the dynamic object can be largely captured by our dynamic token detection module(Sec.4.1.2). The targeting process is categorized as coarse-grained, which is aligned with our idea. (2) When the end effector is touching the object, the velocity is relative lower and would be categorized as fine-grained action, enabling precise action generation.
>
> We provide empirical validation on VLA-Arena dynamic benchmarks:
>
> |VLA-arena  | Success Rate ||SpeedUp|
> | -- | :---: | :---:|:--:|
> | | Dynamic Distractors|Dynamic obstacles | Average
> | OFT | 56.8% | 59.6% | 1.00 $\times$ |
> | ours | 56.0% | 59.4% | 1.40 $\times$ |
>
> Results confirm our method maintains comparable success rates while achieving consistent speedup.
>
> For extremely dynamic scenarios (e.g., catching, ball-playing), we admit our current heuristic may default to coarse-grained pruning and may cause action error. However, such tasks remain challenging for most of VLAs at current stage. We admit this flaw and we are considering improving the action mode detction by relative speed calculation or integrating lightweight learning-based mode classifiers.
>
> ## Re to W3:
>
> In our paper, we also includes LIBERO-plus result in Sec.7.6. We understand the reviewers' concern and add the evaluation in SimplerEnv with CogACT model. The result is listed above in response to W1.
>
> ## Re to W4:
> Thanks to the VLA-cache's open-source work, we deploy the method on our Flexiv robot arm and train with the same recipe as ours. On the same four tasks, the result is as follows:
>
> The vla-cache maintains the same success rate levels as ours but it achieves lower speedup which is same as the simulation results. The reason is that it caches the redundant tokens which is of no use in tasks rather than prune it. This cause a limit drop in computation.
>
> |Method|Pick&Place|TransferTube|PickUpCup|MultiCubeTask|Avg Latency (ms)|Avg Speedup|
> | :-| :-: | :-: | :-: | :-: | :---: |:-:|
> | OpenVLA-OFT | 96.7% | 85.0% | 91.7%|95.0%|187.7|1.00×|
> | Ours | 96.7%| 82.0%| 90.0% | 95.0%|110.2 | 1.70× |
> | VLA-Cache | 95.8% | 82.3%| 90.0% | 94.7% |126.7 |1.48× |

---

> > ### Author Rebuttal · Reviewer_jTvd · 2026-04-03
> >
> > I have same concern with in terms of novelty of method, parameter sensitivity, fairness of comparison. Therefore, I still keep the rating.

---

> > > ### Author Response · Authors · 2026-04-03
> > >
> > > We would like to clarify that SpecPrune-VLA is a principled framework driven by two VLA-specific insights that distinguish it from prior works (e.g., FastV, EfficientVLA):
> > >
> > > 1. From Local Heuristics to Global Semantics: Existing methods rely on local, shallow features to predict token importance. We empirically find (Sec 3.1) that these layers often fail to capture the feature of the whole model.
> > >
> > > Contribution: We introduce Action-Level Static Pruning, leveraging Spatial-Temporal Consistency to use global hierarchical context for guiding pruning. This ensures that task-relevant information is preserved throughout the execution.
> > >
> > > 2. Entropy-Weighted Scoring (derived from Hierarchical Attention Patterns): Treating all layers uniformly ignores the functional hierarchy of VLA models.
> > >
> > > Contribution: The Entropy-Weighted Scoring adaptively weighs token importance based on layer-specific confidence (attention entropy). This explicitly addresses the shift from shallow visual features to deep, action-centric semantics.
> > >
> > > To our knowledge, we are the first to pruning from the global perspective of the model. And in A.1., ablation study on **GLOBAL ATTENTION REUSE** and **ENTROPY-BASED LAYER WEIGHT** show the importance of our proposed component.
> > >
> > > ## Parameter sensitivity
> > >
> > > **The threshold $\tau$ for Dynamic Token Selection:** Filters low‑change patches to avoid noise. Since we select the top-$K_{dynamic}$ most dissimilar patches among those below $\tau$ , the exact value is not sensitive. Experiments shows performance remains stable within a range of $\tau \in$[0.9,0.99] in simulation and $\tau \in$[0.6,0.9] in real world, confirming robustness.
> > >
> > > **Update Rate $\beta$ in dynamic pruning:** Empirically, in exponential moving average (EMA), update rate is often set to a small value to ensure smooth calculation. In our method, $\beta$ works well in a loose range (0.1 - 0.3). When $\beta$>0.6 overly aggressive updates amplify early-layer noise, causing unimportant tokens to receive high importance scores.
> > >
> > > **Dynamic prune rate $\gamma$**: We initially set $\gamma$ to a relative large value 0.9. And we ablate it with overall prune rate $\alpha$(Sec.7.3) by scaling $\gamma$($\alpha$)=$\alpha \cdot \gamma$.
> > >
> > > **Robustness of our method**: In Fig. 7, Sec.7.5, we show that under different prune rate, our method consistently maintain a high succcess rate across various tasks. It shows that our method isn't sensitive to the main parameters prune rate, it only serves as a trade-off between accuracy and speedup.
> > >
> > > ## Implementation detail of SP-VLA
> > > Below we provide detailed implementation notes for our reproduction of the SP-VLA method, including hyperparameter settings, action type classification logic, buffer mechanism, and the token pruning strategy.
> > >
> > > ## 1. Hyperparameter Settings
> > >
> > > To ensure fairness, we strictly follow the hyperparameter configurations from the original source code for each LIBERO task suite, as shown in the table below:
> > >
> > > |Hyperparameter|libero_spatial|libero_object|libero_goal |libero_10|Description|
> > > |-|-|-|-|-|-|
> > > |`z_xy_rate_skip`|0.4| 0.6| 1.2 |1.0|Threshold to determine whether to use the lightweight predictor |
> > > |`z_max_skip`| 0.3|0.5| 0.3|0.3 |Maximum absolute Z-axis displacement threshold|
> > > |`z_thre_prune`|0.5 | 0.5 | 0.3|0.3|Z-axis velocity threshold for triggering token pruning|
> > > |`z_min_prune` | 0.9| 0.9| 0.7| 0.85| Maximum pruning ratio, controlling the minimum proportion of tokens to retain |
> > > | `step_skip`| 1|2|2| 2|Step interval for checking whether the lightweight predictor can be used |
> > >
> > > ## 2. Classification Logic for Deliberative vs. Intuitive Actions
> > >
> > > SP-VLA classify actions based on their displacement characteristics in the Z-axis and XY-plane:
> > >
> > > - **Deliberative Actions (using the full VLA model)**:
> > >   - Z-axis displacement is relatively large (significant vertical movement)
> > >
> > > - **Intuitive Actions (using the lightweight predictor)**:
> > >   - Both of the following conditions are met:
> > >     - `max_z / max_xy < z_xy_rate_skip`: Z-axis displacement is small relative to XY-plane displacement
> > >     - `max_z < z_max_skip`: Absolute Z-axis displacement is small
> > >
> > > ## 3. Action Buffer and Calling Strategy
> > >
> > > - **Buffer Size**: `save_action_number = 6`.
> > > - **Condition for Calling the Lightweight Predictor**: The lightweight predictor is only used if at least 4 out of the last 6 actions were generated by the VLA model.
> > >
> > > ## 4. Spatio-Semantic Dual-Aware Token Pruning
> > >
> > > Token pruning is implemented with the following pipeline:
> > >
> > > 1. Extract the Q and K matrices from the last attention layer of the SigLIP encoder.
> > > 2. Compute attention scores to evaluate the semantic importance of each token.
> > > 3. Dynamically adjust the pruning threshold based on the current Z-axis velocity.
> > > 4. Apply the Canny edge detector to extract spatial structure information from the image.
> > > 5. Merge semantic importance and spatial importance to retain the most critical tokens.
> > >
> > > We would happy to update the results if the official code support OpenVLA-OFT.

---

### Official Review · Reviewer_MX34 · 2026-03-10

**Soundness:** 3
**Presentation:** 3
**Significance:** 3
**Originality:** 3
**Overall Recommendation:** 4
**Confidence:** 2

**Summary:**

This paper proposes a pruning method for Vision Language Action (VLA) models that reduces inference time while minimizing performance degradation. The approach comprises action level static pruning and layer level dynamic pruning, with overall pruning aggressiveness controlled by an action aware controller. Notably, the method requires no additional training, and extensive experiments demonstrate its effectiveness.

**Compliance With Llm Reviewing Policy:**

Affirmed.

**Final Justification:**

The authors’ rebuttal adequately addresses my main concerns by providing clearer justification for the hyperparameter choices and additional experimental evidence on dynamic scenes and long‑horizon tasks. While some limitations remain, such as the relatively large number of parameters and the reliance on heuristic design choices, these represent reasonable design trade‑offs rather than issues that undermine the core contribution. Overall, the paper is technically sound, and the rebuttal reinforces my original assessment, so I maintain my score and recommendation.

**Key Questions For Authors:**

1. When processing long‑horizon sequences, could error accumulation from previous steps affect the reliability of pruning decisions? If so, could the authors elaborate on methods to mitigate these errors?

**Limitations:**

The current evaluation primarily assumes environments where the robot arm moves while external objects remain static. It would be valuable to investigate whether the proposed pruning strategy remains effective in environments where the external scene changes rapidly.

**Strengths And Weaknesses:**

**Strengths.**
- The paper is well structured.
- The proposed method is well motivated, particularly for VLA settings where fast decision making and reduced inference latency are critical. The approach is grounded in an observation based on token level attention scores.
- The experimental evaluation is comprehensive, covering not only a wide range of benchmark tasks but also real world robotic deployments, which supports the practicality of the proposed method.

**Weaknesses.**
- Although the paper provides detailed explanations for selecting hyperparameters, their quantity is relatively large, and several of them appear to be chosen empirically rather than through a systematic procedure.
- The controller determines coarse and fine action phases solely based on the end effector velocity, without considering external environmental cues. Such a heuristic may fail in scenarios where velocity alone is not a reliable indicator of action granularity.

---

> ### Author Rebuttal · Authors · 2026-03-29
>
> ## Re to W1:
> We appreciate the reviewer's insightful comment regarding the hyperparameters. We would like to clarify that while our method involves several parameters, their selection is driven by data-driven physical insights and quantifiable metrics rather than arbitrary empirical tuning. Specifically: The main parameters top-Ks are selected based on the goal of maximizing the Recall of important tokens(Sec.3.2) between consecutive inference. The prune ratio indicates how much part of image should be retained, serving as the trade-off between success rate and speedup. The velosity thresholds represent the physical movement pattern, rather than purely tuning. Also, we provide the experiment on a new model CogACT[1] in a different simulation benchmark SimplerEnv to show the generalization of our methods and setings.
>
> |SimplerEnv|||**success rate**|||\||SpeedUp|
> |:-:|:-:|:-:|:-:|:-:|:-:|:-:|:-:|
> |Method|PickCokeCan|MoveNear|Open/CloseDrawer|AppleInDrawer|Average|\||Average|
> |CogACT|91.3%|85.0%|71.8%|50.8%|74.7%|\||1.00 $\times$|
> |Ours|91.3%|83.8%|71.6%|50.6%|74.3%|\||1.30 $\times$|
> |Ours+cache|91.7%|83.8%|71.2%|50.0%|74.1%|\||1.42 $\times$|
>
> In SimplerEnv, four visual matching tasks, our method alone achieves 1.30 $\times$ speedup. Furthermore, CogACT has a diffusion acion expert. Our method is orthogonal to accelerartion technique in diffusion model such as feature caching. When combining with uniform feature caching (interval=5), we achieve 1.42 $\times$ speedup with negaligible loss in success rate.
>
> ## Re to W2:
> From the perspective of robot, we introduce the action controller to make the task process a close-loop process, which is close to human nature and precise robot manipulation. And in the case of external environmental changing(e.g. dynamic objects), our dynamic detection module(Sec.4.1.2) will take them into account. Furthermore, we supplement the dynamic object scene experiment on VLA-Arena[2]:
>
> |VLA-arena  | Success Rate ||SpeedUp|
> |:--: | :---: | :---:|:--:|
> |  | Dynamic Distractors|Dynamic obstacles | Average
> | OpenVLA-OFT | 56.8% | 59.6% | 1.00 $\times$ |
> | ours | 56.0% | 59.4% | 1.40 $\times$ |
>
> The result shows that our method achieves same level of success rate in dynamic scene, which means it can capture the dynamic elements.
>
> ## Re to Q1:
> We sincerely thank the reviewer for this insightful question regarding error accumulation in long-horizon sequence processing. This is indeed a critical concern for robot manipulation tasks, and we appreciate the opportunity to elaborate on our approach. First, as the VLA model inference is genrally a prefill-only process (every time a new sequence of tokens serves as inputs) and we include locally-important and dynamic tokens in the current inference, there is an upper-bound for our token out-of-date rate, $\frac{V_{global}}{V_{retain}}$. Secondly, as mentioned in our motivation, we point out the spatial-temporal consistency in VLA tasks and elaberately select the globally-important tokens from the last inference, the error is relatively small. In our paper we show that over 75~85% tokens are overlapped.
>
> Furthermore, we supplement a long-horizon tasks set experiment on VLA-Arena:
>
> ||||
> | --- | :---: | :---:|
> | | success rate| SpeedUp |
> | OFT | 80.0% | 1.00 $\times$ |
> | ours | 79.4% | 1.42 $\times$ |
>
>  **VLA-arena Long horizon L0**
>
> ***Future Directions:*** We agree that further refining the estimation of globally-important tokens could enhance long-horizon reliability. In future work, we plan to integrate historical trajectory information with current end-effector velocity to build a predictive model for token global importance, enabling more adaptive and context-aware pruning.
>
> We would explicitly add the above result and discussion for dynamic scene into the manuscript. Thank you again for your valuable review.
>
> [1]Li, Qixiu, et al. "Cogact: A foundational vision-language-action model for synergizing cognition and action in robotic manipulation." arXiv preprint arXiv:2411.19650 (2024).
>
> [2]Zhang, Borong et al. “VLA-Arena: An Open-Source Framework for Benchmarking Vision-Language-Action Models.” ArXiv abs/2512.22539 (2025): n. pag.

---

> > ### Author Rebuttal · Reviewer_MX34 · 2026-04-02
> >
> > My concerns and questions are sufficiently addressed by the rebuttal. While some limitations remain, they reflect design trade-offs rather than unresolved issues that undermine the core contribution, and I therefore maintain my original score.

---

> > > ### Author Response · Authors · 2026-04-02
> > >
> > > We thank the reviewer for the positive acknowledgmen. We are glad that our rebuttal has fully addressed the concerns.

---

### Decision · Program_Chairs · 2026-04-30

**Decision:**

Accept (regular)

**Comment:**

This paper proposes a training-free pruning framework for Vision-Language-Action models, combining static and dynamic pruning to reduce latency while preserving performance. The method is well-motivated and shows consistent improvements across benchmarks and real-world robotic tasks.

Reviewers agree on the importance and practical relevance of the problem, and find the framework coherent with solid empirical support. However, concerns remain: the approach relies heavily on heuristic design and hyperparameters with limited theoretical grounding, raising questions about robustness and generalization. The evaluation, while fairly comprehensive, could be strengthened with broader coverage, clearer comparisons, and deeper efficiency analysis. Novelty and clarity of core contributions are also somewhat limited.

The rebuttal provides additional experiments and clarifications, addressing several empirical concerns, though higher-level issues (e.g., heuristic nature and novelty) are only partially resolved. Reviewer opinions are mixed but lean positive.

Overall, despite limitations in theoretical justification and novelty, the method demonstrates consistent practical gains, and the rebuttal strengthens confidence in the results. I therefore recommend Weak Accept.